# Argoverse 2: Next Generation Datasets for Self-Driving Perception and Forecasting

**Benjamin Wilson**[#][*][†]**, William Qi**[#][*]**, Tanmay Agarwal**[#][*]**, John Lambert**[*][†]**, Jagjeet Singh**[*],

**Siddhesh Khandelwal**[‡]**, Bowen Pan**[*][§]**, Ratnesh Kumar**[*]**, Andrew Hartnett**[*],

**Jhony Kaesemodel Pontes**[*]**, Deva Ramanan**[*][¶]**, Peter Carr**[*]**, James Hays**[*][†]

Argo AI[*], Georgia Tech[†], UBC[‡], MIT[§], CMU[¶]

{bwilson, wqi, tagarwal, jlambert, jsingh, bpan, ratneshk, ahartnett, jpontes, dramanan, pcarr, jhays}@argo.ai, {skhandel}@cs.ubc.ca

## Abstract

We introduce Argoverse 2 (AV2) — a collection of three datasets for perception and forecasting research in the self-driving domain. The annotated *Sensor Dataset* contains 1,000 sequences of multimodal data, encompassing high-resolution imagery from seven ring cameras, and two stereo cameras in addition to lidar point clouds, and 6-DOF map-aligned pose. Sequences contain 3D cuboid annotations for 26 object categories, all of which are sufficiently-sampled to support training and evaluation of 3D perception models. The *Lidar Dataset* contains 20,000 sequences of unlabeled lidar point clouds and map-aligned pose. This dataset is the largest ever collection of lidar sensor data and supports self-supervised learning and the emerging task of point cloud forecasting. Finally, the *Motion Forecasting Dataset* contains 250,000 scenarios mined for interesting and challenging interactions between the autonomous vehicle and other actors in each local scene. Models are tasked with the prediction of future motion for "scored actors" in each scenario and are provided with track histories that capture object location, heading, velocity, and category. In all three datasets, each scenario contains its own *HD Map* with 3D lane and crosswalk geometry — sourced from data captured in six distinct cities. We believe these datasets will support new and existing machine learning research problems in ways that existing datasets do not. All datasets are released under the CC BY-NC-SA 4.0 license.

## 1 Introduction

In order to achieve the goal of safe, reliable autonomous driving a litany of machine learning tasks must be addressed, from stereo depth estimation to motion forecasting to 3D object detection. In recent years, numerous high quality self-driving datasets have been released to support research into these and other important machine learning tasks. Many datasets are annotated "sensor" datasets [4, 39, 34, 35, 21, 28, 16, 12, 36, 31] in the spirit of the influential KITTI dataset [15]. The Argoverse 3D Tracking dataset [6] was the first such dataset with "HD maps" — maps containing lane-level geometry. Also influential are self-driving "motion prediction" datasets [11, 19, 29, 4, 45] — containing abstracted object tracks instead of raw sensor data — of which the Argoverse Motion Forecasting dataset [6] was the first.

In the last two years, the Argoverse team has hosted six competitions on 3D tracking, stereo depth estimation, and motion forecasting. We maintain evaluation servers and leaderboards for these tasks, as well as 3D detection. The leaderboards collectively contain thousands of submissions from four

---

[#]equal contribution

35th Conference on Neural Information Processing Systems (NeurIPS 2021) Track on Datasets and Benchmarks.

hundred teams[1]. We also maintain the Argoverse API and have addressed more than one hundred issues[2]. From these experiences we have formed the following guiding principles to guide the creation of the next iteration of Argoverse datasets.

1. **Bigger isn't always better.** Self-driving vehicles capture a flood of sensor data which is logistically difficult to work with. Sensor datasets are several terabytes in size, even when compressed. If standard benchmarks grow further, we risk alienating much of the academic community and leaving progress to well-resourced industry groups. *For this reason, we match but do not exceed the scale of sensor data in nuScenes [4] and Waymo Open [39].*

2. **Make every instance count.** Much of driving is boring. Datasets should focus on the difficult, interesting scenarios where current forecasting and perception systems struggle. *Therefore we mine for especially crowded, dynamic, and kinematically unusual scenarios.*

3. **Diversity matters.** Training on data from wintertime Detroit is not sufficient for detecting objects in Miami — Miami has 15 times the frequency of motorcycles and mopeds. Behaviors differ as well, so learned pedestrian motion behavior might not generalize. *Accordingly, each of our datasets are drawn from six diverse cities — Austin, Detroit, Miami, Palo Alto, Pittsburgh, and Washington D.C. — and different seasons, as well, from snowy to sunny.*

4. **Map the world.** HD maps are powerful priors for perception and forecasting. Learning-based methods that found clever ways to encode map information [27] performed well in Argoverse competitions. *For this reason, we augment our HD map representation with 3D lane geometry, paint markings, crosswalks, higher resolution ground height, and more.*

5. **Self-supervise.** Other machine learning domains have seen enormous success from self-supervised learning in recent years. Large-scale lidar data from dynamic scenes, paired with HD maps, could lead to better representations than current supervised approaches. *For this reason, we build the largest dataset of lidar sensor data.*

6. **Fight the heavy tail.** Passenger vehicles are common, and thus we can assess our forecasting and detection accuracy for cars. However, with existing datasets, we cannot assess forecasting accuracy for buses and motorcycles with their distinct behaviors, nor can we evaluate stroller and wheel chair detection. *Thus we introduce the largest taxonomy to date for sensor and forecasting datasets, and we ensure enough samples of rare objects to train and evaluate models.*

With these guidelines in mind we built the three Argoverse 2 (AV2) datasets. Below, we highlight some of their contributions.

1. The 1,000 scenario *Sensor dataset* has the largest self-driving taxonomy to date – 30 categories. 26 categories contain at least 6,000 cuboids to enable diverse taxonomy training and testing. The dataset also has stereo imagery, unlike recent self-driving datasets.

2. The 20,000 scenario *Lidar dataset* is the largest dataset for self-supervised learning on lidar. The only similar dataset, concurrently developed ONCE [31], does not have HD maps.

3. The 250,000 scenario *Motion Forecasting Dataset* has the largest taxonomy – 5 types of dynamic actors and 5 types of static actors – and covers the largest mapped area of any such dataset.

We believe these datasets will support research into problems such as 3D detection, 3D tracking, monocular and stereo depth estimation, motion forecasting, visual odometry, pose estimation, lane detection, map automation, self-supervised learning, structure from motion, scene flow, optical flow, time to contact estimation, and point cloud forecasting.

## 2   Related Work

The last few years have seen rapid progress in self-driving perception and forecasting research, catalyzed by many high quality datasets.

**Sensor datasets and 3D Object Detection and Tracking.** New sensor datasets for 3D object detection [4, 39, 34, 35, 21, 28, 16, 12, 36, 31] have led to influential detection methods such as anchor-based approaches like PointPillars [23], and more recent anchor-free approaches such as

---

[1]This count includes private submissions not posted to the public leaderboards.
[2]https://github.com/argoai/argoverse-api

AFDet [14] and CenterPoint [44]. These methods have led to dramatic accuracy improvements on all datasets. In turn, these improvements have made isolation of object-specific point clouds possible, which has proven invaluable for offboard detection and tracking [37], and for simulation [8], which previously required human-annotated 3D bounding boxes [30]. New approaches explore alternate point cloud representations, such as range images [5, 2, 40]. Streaming perception [25, 18] introduces a paradigm to explore the tradeoff between accuracy and latency. A detailed comparison between the AV2 *Sensor Dataset* and recent 3D object detection datasets is provided in Table 1.

**Motion Forecasting.** For motion forecasting, the progress has been just as significant. A transition to attention-based methods [24, 33, 32] has led to a variety of new vector-based representations for map and trajectory data [13, 27]. New datasets have also paved the way for new algorithms, with nuScenes [4], Lyft L5 [19], and the Waymo Open Motion Dataset [11] all releasing lane graphs after they proved to be essential in Argoverse 1 [6]. Lyft also introduced traffic/speed control data, while Waymo added crosswalk polygons, lane boundaries (with marking type), speed limits, and stop signs to the map. More recently, Yandex has released the Shifts [29] dataset, which is the largest (by scenario hours) collection of forecasting data available to date. Together, these datasets have enabled exploration of multi-actor, long-range motion forecasting leveraging both static and dynamic maps.

Following upon the success of Argoverse 1.1, we position AV2 as a large-scale repository of high-quality motion forecasting scenarios - with guarantees on data frequency (exactly 10 Hz) and diversity (>2000 km of unique roadways covered across 6 cities). This is in contrast to nuScenes (reports data at just 2 Hz) and Lyft (collected on a single 10 km segment of road), but is complementary to Waymo Open Motion Dataset (employs a similar approach for scenario mining and data configuration). Complementary datasets are essential for these safety critical problems as they provide opportunities to evaluate generalization and explore transfer learning. To improve ease of use, we have also designed AV2 to be widely accessible both in terms of data size and format — a detailed comparison vs. other recent forecasting datasets is provided in Table 2.

**Broader Problems of Perception for Self-Driving.** Aside from the tasks of object detection and motion forecasting, new, large-scale sensor datasets for self-driving present opportunities to explore dozens of new problems for perception, especially those that can be potentially solved via self-supervision. A number of new problems have been recently proposed; real-time 3D semantic segmentation in video has received attention thanks to SemanticKITTI [1]. HD map automation [46, 26] has received additional attention, along with 3D scene flow and pixel-level scene simulation [43, 8]. Datasets exist with unique modalities such as thermal imagery [10, 9]. Our new *Lidar Dataset* enables large-scale self-supervised training of new approaches for freespace forecasting [20] or point cloud forecasting [41, 42].

## 3 The Argoverse 2 Datasets

### 3.1 Sensor Dataset

The *Argoverse 2 Sensor Dataset* is the successor to the *Argoverse 1 3D Tracking Dataset*. AV2 is larger, with 1,000 scenes, up from 113 in Argoverse 1, but each AV2 scene is also richer – there are 23x as many non-vehicle, non-pedestrian cuboids in AV2. The constituent scenarios in the Argoverse 2 Sensor Dataset were manually selected by the authors to contain crowded scenes with under-represented objects, noteworthy weather, and interesting behaviors, e.g. cut ins and jaywalking. Each scenario is fifteen seconds. Table 1 compares the AV2 Sensor Dataset with a selection of self-driving datasets. Figures 1, 2, and 3 plot how the scenarios of AV2 compare favorably to other datasets in terms of annotation range, object diversity, object density, and scene dynamism.

The most similar sensor dataset to ours is the highly influential nuScenes [4] – both datasets have 1,000 scenarios and HD maps, although Argoverse is unique in having ground height maps. nuScenes contains radar data while the AV2 contains stereo imagery. nuScenes has a large taxonomy – twenty-three object categories of which ten have suitable data for training and evaluation. Our dataset contains thirty object categories of which twenty-six are well sampled enough for training and evaluation. nuScenes spans two cities while our proposed dataset spans six.

**Sensor Suite.** Lidar sweeps are collected at 10 Hz, along with 20 fps imagery from 7 cameras positioned to provide a fully panoramic field of view. In addition, camera intrinsics, extrinsics and 6-DOF ego-vehicle pose in a global coordinate system are provided. Lidar returns are captured by

Table 1: Comparison of the Argoverse 2 *Sensor* and *Lidar* datasets with other sensor datasets.

| Name | # Scenes | Cities | Lidar? | # Cameras | Stereo | HD Maps? | # Classes | # Evaluated Classes |
|------|----------|--------|--------|-----------|--------|----------|-----------|---------------------|
| Argoverse 1 [6] | 113 | 2 | ✓ | 7 | ✓ | ✓ | 15 | 3 |
| KITTI [15] | 22 | 1 | ✓ | 2 | ✓ | | 3 | 3 |
| nuScenes [4] | 1,000 | 2 | ✓ | 6 | | ✓ | 23 | 10 |
| ONCE [31] | 581 | – | ✓ | 7 | | | 5 | 3 |
| Waymo Open [39] | 1,150 | 3 | ✓ | 5 | | | 4 | 4 |
| Argoverse 2 Sensor | 1,000 | 6 | ✓ | 9 | ✓ | ✓ | 30 | 26 |
| Argoverse 2 Lidar | 20,000 | 6 | ✓ | - | | ✓ | - | - |

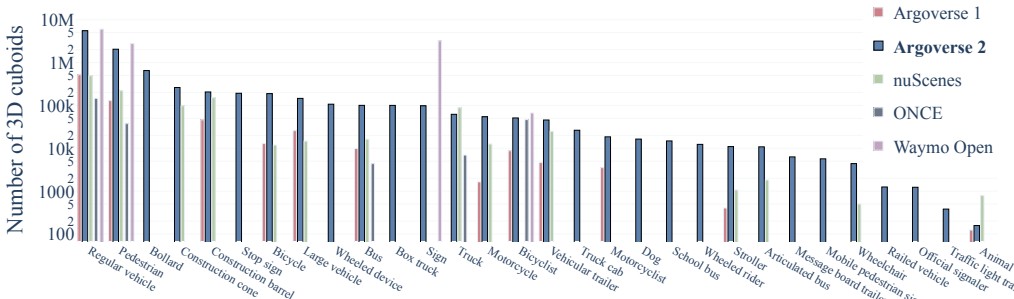

Figure 1: Number of annotated 3D cuboids per category for Argoverse 1 *3D Tracking*, Argoverse 2 *Sensor Dataset*, nuScenes, ONCE, and Waymo Open. The nuScenes annotation rate is 2 Hz, compared to 10 Hz for Argoverse, but that does not account for the relative increase in object diversity in Argoverse 2.

two 32-beam lidars, spinning at 10 Hz in the same direction, but separated in orientation by 180°. The cameras trigger in-sync with both lidars, leading to a 20 Hz frame-rate. The seven global shutter cameras are synchronized to the lidar to have their exposure centered on the lidar sweeping through their fields of view. In the Supplementary Material, we provide a a schematic figure illustrating the car sensor suite and its coordinate frames.

**Lidar synchronization accuracy.** In AV2, we improve the synchronization of cameras and lidars significantly over Argoverse 1. Our synchronization accuracy is within $[-1.39, 1.39]$ ms, which compares favorably to the Waymo Open Dataset, which is reported as $[-6, 7]$ ms [39].

**Annotations.** The AV2 Sensor Dataset contains 10 Hz 3D cuboid annotations for objects within our 30 class taxonomy (Figure 1). Cuboids have track identifiers that are consistent over time for the

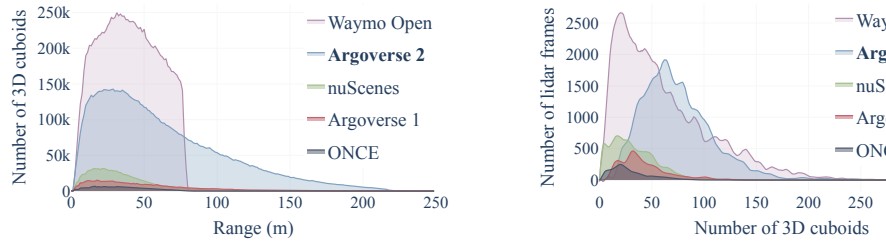

Figure 2: **Left:** Number of annotated 3D cuboids by range in the Argoverse 2 *Sensor Dataset*. About 14% of the Argoverse 2 cuboids are beyond 75 m – Waymo Open, nuScenes, and ONCE have less than 1%. **Right:** Number of 3D cuboids per lidar frame. Argoverse 2 has an average of 75 3D cuboids per lidar frame – Waymo Open has an average of 61, nuScenes 33, and ONCE 30.

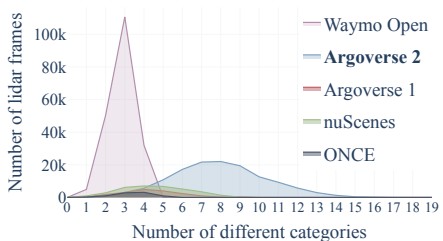 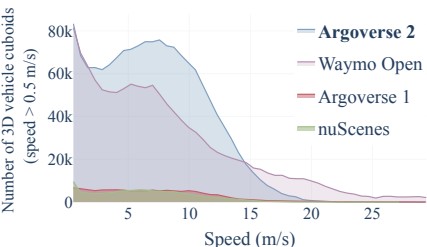

Figure 3: **Left:** Number of annotated categories per lidar frame in the Argoverse 2 *Sensor Dataset*. Per scene, Argoverse 2 is about $2\times$ more diverse than Argoverse 1 and $2.3\times$ more diverse than Waymo Open. **Right:** Speed distribution for the vehicle category. We consider only moving vehicles with speeds greater than 0.5 m/s. Argoverse 2 has about $1.3\times$ more moving vehicles than Waymo Open. About 28% of the vehicles in Argoverse 2 are moving with an average speed of 7.27 m/s. We did not compare against the ONCE dataset because it does not provide tracking information for the 3D cuboids.

same object instance. Objects are annotated if they are within the "region of interest" (ROI) – within five meters of the mapped "driveable" area.

**Privacy.** All faces and license plates, whether inside vehicles or outside of the driveable area, are blurred extensively to preserve privacy.

**Sensor Dataset splits**. We randomly partition the dataset with train, validation, and test splits of 700, 150, and 150 scenarios, respectively.

### 3.2 Lidar Dataset

The *Argoverse 2 Lidar Dataset* is intended to support research into self-supervised learning in the lidar domain as well as point cloud forecasting [41, 42]. Because lidar data is more compact than the full sensor suite, we can include far more scenarios – 20,000 instead of 1,000 – for roughly the same space budget. The AV2 Lidar Dataset is mined with the same criteria as the Forecasting Dataset (Section 3.3.2) to ensure that each scene is interesting. While the Lidar Dataset does not have 3D object annotations, each scenario carries an HD map with rich, 3D information about the scene.

Our dataset is the largest such collection to date with 20,000 thirty second sequences. The only similar dataset, concurrently released ONCE [31], contains $1\,\mathrm{M}$ lidar frames compared to $6\,\mathrm{M}$ lidar frames in ours. Our dataset is sampled at $10\,\mathrm{Hz}$ instead of $2\,\mathrm{Hz}$, as in ONCE, making our dataset more suitable for point cloud forecasting or self-supervision tasks where point cloud evolution over time is important.

**Lidar Dataset splits**. We randomly partition the dataset with train, validation, and test splits of 16,000, 2,000, and 2,000 scenarios, respectively.

### 3.3 Motion Forecasting Dataset

Motion forecasting addresses the problem of predicting future states (or occupancy maps) for dynamic actors within a local environment. Some examples of relevant actors for autonomous driving include: vehicles (both parked and moving), pedestrians, cyclists, scooters, and pets. Predicted futures generated by a forecasting system are consumed as the primary inputs in motion planning, which conditions trajectory selection on such forecasts. Generating these forecasts presents a complex, multi-modal problem involving many diverse, partially-observed, and socially interacting agents. However, by taking advantage of the ability to "self-label" data using observed ground truth futures, motion forecasting becomes an ideal domain for application of machine learning.

Building upon the success of Argoverse 1, the Argoverse 2 Motion Forecasting dataset provides an updated set of prediction scenarios collected from a self-driving fleet. The design decisions

---

[3]True if interesting scenarios/actors are mined *after data collection*, instead of taking all/random samples.
[4]As retrieved on Aug. 27, 2021.

Table 2: Comparison between the Argoverse 2 Motion Forecasting dataset and other recent motion forecasting datasets. Hyphens "-" indicate that attributes are either not applicable, or not available.

| | ARGOVERSE [6] | INTER [45] | LYFT [19] | WAYMO [11] | NUSCENES [4] | YANDEX [29] | OURS |
|---|---|---|---|---|---|---|---|
| # SCENARIOS | 324k | - | 170k | 104k | 41k | 600k | 250k |
| # UNIQUE TRACKS | 11.7M | 40k | 53.4M | 7.6M | - | 17.4M | 13.9M |
| AVERAGE TRACK LENGTH | 2.48 s | 19.8 s | 1.8 s | 7.04 s | - | - | 5.16 s |
| TOTAL TIME | 320 h | 16.5 h | 1118 h | 574 h | 5.5 h | 1667 h | 763 h |
| SCENARIO DURATION | 5 s | - | 25 s | 9.1 s | 8 s | 10 s | 11 s |
| TEST FORECAST HORIZON | 3 s | 3 s | 5 s | 8 s | 6 s | 5 s | 6 s |
| SAMPLING RATE | 10 Hz | 10 Hz | 10 Hz | 10 Hz | 2 Hz | 5 Hz | 10 Hz |
| # CITIES | 2 | 6 | 1 | 6 | 2 | 6 | 6 |
| UNIQUE ROADWAYS | 290 km | 2 km | 10 km | 1750 km | - | - | 2220 km |
| AVG. # TRACKS PER SCENARIO | 50 | - | 79 | - | 75 | 29 | 73 |
| # EVALUATED OBJECT CATEGORIES | 1 | 1 | 3 | 3 | 1 | 2 | 5 |
| MULTI-AGENT EVALUATION | × | ✓ | ✓ | ✓ | × | ✓ | ✓ |
| MINED FOR INTERESTINGNESS[3] | ✓ | × | - | ✓ | × | × | ✓ |
| VECTOR MAP | ✓ | × | × | ✓ | ✓ | × | ✓ |
| DOWNLOAD SIZE | 4.8 GB | - | 22 GB | 1.4 TB | 48 GB | 120 GB | 32 GB |
| # PUBLIC LEADERBOARD ENTRIES[4] | 194 | - | 935 | 23 | 18 | 3 | - |

enumerated below capture the collective lessons learned from both our internal research/development, as well as feedback from more than 2,700 submissions by nearly 260 unique teams[5] across 3 competitions [38].

1. **Motion forecasting is a safety critical system in a long-tailed domain.** Consequently, our dataset is biased towards diverse and interesting scenarios containing different types of focal agents (see section 3.3.2). Our goal is to encourage the development of methods that ensure safety during tail events, rather than to optimize the expected performance on "easy miles".

2. **There is a "Goldilocks zone" of task difficulty.** Performance on the Argoverse 1 test set has begun to plateau, as shown in the supplemental. Argoverse 2 is designed to increase prediction difficulty incrementally, spurring productive focused research for the next few years. These changes are intended to incentivize methods that perform well on extended forecast horizons (3 s -> 6 s), handle multiple types of dynamic objects (1 -> 5), and ensure safety in scenarios from the long tail. Future Argoverse releases could continue to increase the problem difficulty by reducing observation windows and increasing forecasting horizons.

3. **Usability matters.** Argoverse 1 benefited from a large and active research community - in large part due to the simplicity of setup and usage. Consequently, we took care to ensure that existing Argoverse models can be easily ported to run on 2. In particular, we have prioritized intuitive access to map elements, encouraging methods which use the lane graph as a strong prior. To improve training and generalization, all poses have also been interpolated and resampled at exactly 10 Hz (Argoverse 1 was approximate). The new dataset includes fewer, longer, and more complex scenarios; this ensures that total dataset size remains large enough to train complex models but small enough to be readily accessible.

### 3.3.1 Data Representation

The dataset consists of 250,000 non-overlapping scenarios (80/10/10 train/val/test random splits) mined from six unique urban driving environments in the United States. It contains a total of 10 object types, with 5 from each of the dynamic and static categories (see Figure 4). Each scenario includes a local vector map and 11 s (10 Hz) of trajectory data (2D position, velocity, and orientation) for all tracks observed by the ego-vehicle in the local environment. The first 5 s of each scenario is denoted as the *observed* window, while the subsequent 6 s is denoted as the *forecasted* horizon.

Within each scenario, we mark a single track as the "focal agent". Focal tracks are guaranteed to be fully observed throughout the duration of the scenario and have been specifically selected to maximize interesting interactions with map features and other nearby actors (see Section 3.3.2). To evaluate multi-agent forecasting, we also mark a subset of tracks as "scored actors" (as shown in Figure 5), with guarantees for scenario relevance and minimum data quality.

---

[5]This count includes private submissions not posted to the public leaderboards.

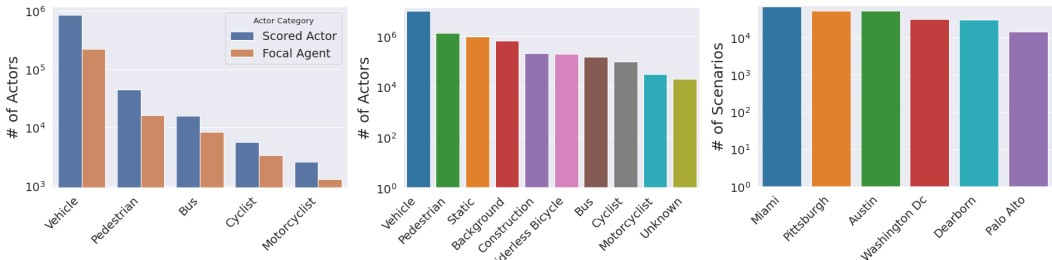

Figure 4: Object type and geographic histograms for the Motion Forecasting Dataset. **Left**: Histogram of object types over the "focal" and "scored" categories. **Center**: Histogram of object types over all tracks present in the dataset. The fine grained distinctions between different static object types (e.g. *Construction Cone* vs *Riderless Bicycle*) are unique among forecasting datasets. **Right**: Histogram of metropolitan areas included in the dataset.

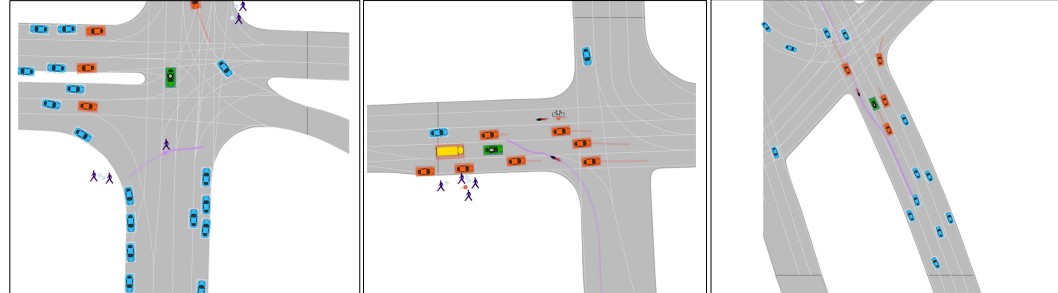

Figure 5: Visualization of a few interesting scenarios from the Motion Forecasting Dataset. The scenarios demonstrate a mix of the various object types (*Vehicle*, *Pedestrian*, *Bus*, *Cyclist*, or *Motorcyclist*). The ego-vehicle is indicated in green, the focal agent is purple, and scored actors are orange. Other un-scored tracks are shown in blue. Object positions are captured at the last timestep of the *observed* history. For visualization purposes the full 5 s history and 6 s future are rendered for the focal agent, while only 1.5 s of future are shown for the other scored actors. **Left** shows a pedestrian crossing in front of the ego-vehicle, while **Center** and **Right** depict a motorcyclist weaving through traffic.

### 3.3.2 Mining Interesting Scenarios

The source data for Argoverse 2 was drawn from fleet logs tagged with annotations consistent with interesting or difficult-to-forecast events. Each log was trimmed to 30 s and run through an *interestingness* scoring module in order to bias data selection towards examples from the long-tail of the natural distribution. We employ heuristics to score each track in the scene across five dimensions: object category, kinematics, map complexity, social context, and relation to the ego-vehicle (details in supplement).

The final scenarios are generated by extracting non-overlapping 11 s windows where at least one candidate track is fully observed for the entire duration. The highest scoring candidate track is denoted as the "focal agent"; all other fully observed tracks within 30 m of the ego-vehicle are denoted as "scored actors". The resulting dataset is diverse, challenging, and still right-sized for widespread use (see the download size in Table 2). In Figure 6, we show that the resulting dataset is significantly more interesting than Argoverse 1.1 and validate our intuition that actors scoring highly in our heuristic module are more challenging to accurately forecast.

### 3.4 HD Maps

Each scenario in the three datasets described above shares the same HD map representation. Each scenario carries its own local map region, similar to the Waymo Open Motion [11] dataset. This is a departure from the original Argoverse datasets in which all scenarios were localized onto two city-scale maps – one for Pittsburgh and one for Miami. In the Supplementary Material, we provide examples. Advantages of per-scenario maps include more efficient queries and their ability to handle

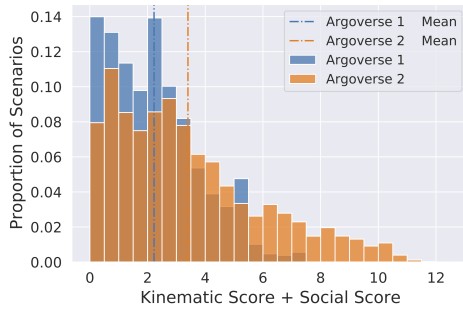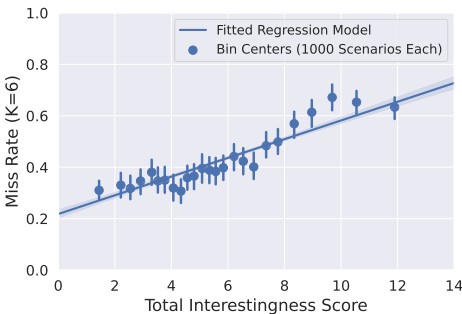

Figure 6: **Left:** Histogram comparing the distribution of interestingness scores assigned to focal agents in both Argoverse 1.1 and 2. **Right:** Plot showing the relationship between total *interestingness score* and prediction difficulty on the Argoverse 2 test split. We evaluate WIMP over each scenario and fit a regression model to the computed miss rate (K=6, 2m threshold).

*map changes*. A particular intersection might be observed multiple times in our datasets, and there could be changes to the lanes, crosswalks, or even ground height in that time.

**Lane graph.** The core feature of the HD map is the lane graph, consisting of a graph $\mathcal{G} = (\mathcal{V}, \mathcal{E})$, where $\mathcal{V}$ are individual lane segments. In the Supplemental Material, we enumerate and define the attributes we provide for each lane segment. Unlike Argoverse 1, we provide the actual 3D lane boundaries, instead of only centerlines. However, our API provides code to quickly infer the centerlines at any desired sampling resolution. Polylines are quantized to $1\,\mathrm{cm}$ resolution. Our representation is richer than nuScenes, which provides lane geometry only in 2D, not 3D.

**Driveable area.** Instead of providing driveable area segmentation in a rasterized format, as we did in Argoverse 1, we release it in a vector format, i.e. as 3D polygons. This offers multiple advantages, chiefly in compression, allowing us to store separate maps for tens of thousands of scenarios, yet the raster format is still easily derivable. The polygon vertices are quantized to $1\,\mathrm{cm}$ resolution.

**Ground surface height.** Only the sensor dataset includes a dense ground surface height map (although other datasets still have sparse 3D height information on polylines). Ground surface height is provided for areas within a $5\,\mathrm{m}$ isocontour of the driveable area boundary, which we define as the *region of interest* (ROI) [6]. We do so because the notion of ground surface height is ill-defined for the interior of buildings and interior of densely constructed city blocks, areas where ground vehicles cannot observe due to occlusion. The raster grid is quantized to a $30\,\mathrm{cm}$ resolution, a higher resolution than the $1\,\mathrm{m}$ resolution in Argoverse 1.

**Area of Local Maps.** Each scenario's local map includes all entities found within a $100\,\mathrm{m}$ dilation in $l_2$-norm from the ego-vehicle trajectory.

## 4 Experiments

Argoverse 2 supports a variety of downstream tasks. In this section we highlight three different learning problems: 3D object detection, point cloud forecasting, and motion forecasting — each supported by the sensor, lidar, and motion forecasting datasets, respectively. First, we illustrate the *challenging* and *diverse* taxonomy within the Argoverse 2 sensor dataset by training a state-of-the-art 3D detection model on our twenty-six evaluation classes including "long-tail" classes such as stroller, wheel chairs, and dogs. Second, we showcase the utility of the Argoverse 2 lidar dataset through *large-scale*, self-supervised learning through the point cloud forecasting task. Lastly, we demonstrate motion forecasting experiments which provide the first baseline for broad taxonomy motion prediction.

### 4.1 3D Object Detection

We provide baseline 3D detection results using a state-of-the-art, anchorless 3D object detection model – CenterPoint [44]. Our CenterPoint implementation takes a point cloud as input and crops it to a $200\,\mathrm{m} \times 200\,\mathrm{m}$ grid with a voxel resolution of $[0.1\,\mathrm{m}, 0.1\,\mathrm{m}]$ in the $xy$ (bird's-eye-view) plane and $0.2\,\mathrm{m}$ in the $z$-axis. To accommodate our larger taxonomy, we include six detection heads

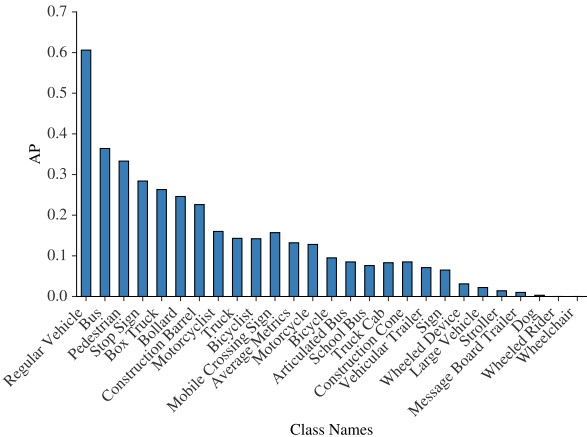

Figure 7: Average precision of our 3D object detection baseline on the *validation* split of the *Sensor Dataset*. Our experiments showcase both our *diverse* taxonomy and *difficult* "long-tail" classes.

to encourage feature specialization. Figure 7 characterizes the performance of our 3D detection baseline using the nuScenes [4] average precision metric. Our large taxonomy allows us to evaluate classes such as "Wheeled Device" (eScooter), "Stroller", "Dog", and "Wheelchair" and we find that performance on these categories with strong baselines is poor despite significant amounts of training data.

## 4.2 Point Cloud Forecasting

We perform point cloud forecasting according to the experimental protocol of SPF2 [42] using the Argoverse 2 Lidar Dataset. Given a sequence of past scene point clouds, a model is required to predict a sequence of future scene point clouds. We take the scene point clouds in the past $1\,\mathrm{s}$ ($10\,\mathrm{Hz}$) in the range image format as input, and then predict the next $1\,\mathrm{s}$ of range images. SPFNet predicts two output maps at each time step – the first output map is the predicted range values, while the second output is a validity mask. Previous point cloud forecasting models were evaluated on smaller KITTI or nuScenes. To explore how the amount of training data affects the performance, we use increasing amounts of data for training the same model architecture, up to the full training set of 16,000 sequences.

**Evaluation.** We use three metrics to evaluate the performance of our forecasting model: *mean IoU*, $l_1$-*norm*, and *Chamfer distance*. The *mean IoU* evaluates the predicted range mask. The $l_1$-*norm* measures the average $l_1$ distance between the pixel sets of predicted range image and the ground-truth image, which are both masked out by the ground-truth range mask. The *Chamfer distance* is obtained by adding up the Chamfer distances in both directions (forward and backward) between the ground-truth point cloud and the predicted scene point cloud which is obtained by back-projecting the predicted range image.

**Results of SPF2 and Discussion.** Table 3 contains the results of our point cloud forecasting experiments. With increasing training data, the performance of the model grows steadily in all three metrics. These results and the works from the self-supervised learning literature [3, 7] indicate that a large amount of training data can make a substantial difference. Another observation is that the Chamfer distances for predictions

Table 3: Results of point cloud forecasting on the *test* split of the *Lidar Dataset*.

| # TRAIN LOGS | 125 | 250 | 500 | 1k | 2k | 4k | 16k |
|---|---|---|---|---|---|---|---|
| MEAN IoU (%) | 55.5 | 63.4 | 61.7 | 65.1 | 68.0 | 68.4 | 70.9 |
| $l_1$-NORM | 13.5 | 12.5 | 11.8 | 9.9 | 8.9 | 8.7 | 7.4 |
| CHAMFER DIST. | 31.1 | 25.9 | 22.4 | 22.9 | 20.5 | 18.2 | 14.0 |

on our dataset are quite a bit higher than predictions on KITTI [42]. We conjecture that this could be due to two reasons: (1) Argoverse 2 Lidar Dataset has a much larger sensing range (above $200\,\mathrm{m}$ versus $120\,\mathrm{m}$ of the KITTI lidar sensor), which tends to significantly increase the value of Chamfer

distance. (2) Argoverse 2 Lidar Dataset has higher proportion of dynamic scenes compared with KITTI Dataset.

### 4.3 Motion Forecasting

We present several forecasting baselines [6] which try to make use of different aspects of the data. Those which are trained using the focal agent only and do not capture any social interaction include: constant velocity, nearest neighbor, and LSTM encoder-decoder models (both with and without map-prior). We also evaluate WIMP [22] as an example of a graph-based attention method that captures social interaction. All hyper-parameters are obtained from the reference implementations.

**Evaluation.** Baseline approaches are evaluated according to standard metrics. Following [6], we use *minADE* and *minFDE* as the metrics; they evaluate the average and endpoint L2 distance respectively, between the best forecasted trajectory and the ground truth. We also use *Miss Rate (MR)* which represents the proportion of test samples where none of the forecasted trajectories were within 2.0 meters of ground truth according to endpoint error. The resulting performance illustrates both the community's progress on the problem and the significant increase in dataset difficulty when compared with Argoverse 1.1.

**Baseline Results.** Table 4 summarizes the results of baselines. For K=1, Argoverse 1 [6] showed that constant velocity model (*minFDE*=7.89) performed better than NN+map(prior) (*minFDE*=8.12), which is not the case here. This further proves that Argoverse 2 is kinematically more diverse and cannot be solved by making constant velocity assumptions. Surprisingly, NN and LSTM variants that make use of map prior perform worse than those who do not, illustrating the scope of improvement in how these baselines leverage the map. For K=6, WIMP significantly outperforms every other baseline. This emphasizes that it is imperative to train expressive models that can leverage map prior and social context along with making diverse predictions. The trends are similar to our past 3 Argoverse Motion Forecasting competitions [38]: Graph-based attention methods (e.g. [22, 27, 32]) continued to dominate the competition, and were nearly twice as accurate as next best baseline (Nearest Neighbor) at K=6. That said, some of the rasterization-based (eg. [17]) methods also showed promising results. Finally, we also evaluated baseline methods in the context of transfer learning and varied object types, the results of which are summarized in supplementary.

Table 4: Performance of motion forecasting baseline methods on vehicle-like (*vehicle*, *bus*, *motorcyclist*) object types. Usage of map prior indicates access to map information whereas usage of social context entails encoding other actors' states in the feature representation. Mining intersection (multimodal) scenarios leads to poor performance at K=1 for all methods. Constant Velocity models have particularly poor performance due to the dataset bias towards kinematically interesting trajectories. Note that modern deep methods such as WIMP still have a miss rate of 0.42 at K=6, indicating the increased difficulty of the Argoverse 2 dataset.

| Model | Map Prior | Social Context | K=1 | | | K=6 | | |
|---|---|---|---|---|---|---|---|---|
| | | | minADE↓ | minFDE↓ | MR↓ | minADE↓ | minFDE↓ | MR↓ |
| Const. Vel. [6] | | | 7.75 | 17.44 | 0.89 | - | - | - |
| NN [6] | | | 4.46 | 11.71 | 0.81 | 2.18 | 4.94 | 0.60 |
| NN [6] | ✓ | | 6.45 | 15.51 | 0.84 | 4.3 | 10.08 | 0.78 |
| LSTM [6] | | | 3.05 | 8.28 | 0.85 | - | - | - |
| LSTM [6] | ✓ | | 5.07 | 12.71 | 0.9 | 3.73 | 9.09 | 0.85 |
| WIMP [22] | ✓ | ✓ | **3.09** | **7.71** | **0.84** | **1.47** | **2.90** | **0.42** |

## 5 Conclusion

**Discussion.** In this work, we have introduced three new datasets that constitute Argoverse 2. We provide baseline explorations for two tasks – point cloud forecasting and motion forecasting. Our datasets provide new opportunities for many other tasks. We believe our datasets compare favorably to existing datasets, with HD maps, rich taxonomies, geographic diversity, and interesting scenes.

**Limitations.** As in any human annotated dataset, there is label noise, although we seek to minimize it before release. 3D bounding boxes of objects are not included in the motion forecasting dataset, but one can make reasonable assumptions about the object extent given the object type. The motion forecasting dataset also has imperfect tracking, consistent with state-of-the-art 3D trackers.

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
