# Supplemental Material for Argoverse 2: Next Generation Datasets for Self-driving Perception and Forecasting

**Benjamin Wilson**[#][*][†]**, William Qi**[#][*]**, Tanmay Agarwal**[#][*]**, John Lambert**[*][†]**, Jagjeet Singh**[*]**,**

**Siddhesh Khandelwal**[‡]**, Bowen Pan**[*][§]**, Ratnesh Kumar**[*]**, Andrew Hartnett**[*]**,**

**Jhony Kaesemodel Pontes**[*]**, Deva Ramanan**[*][¶]**, Peter Carr**[*]**, James Hays**[*][†]

Argo AI[*], Georgia Tech[†], UBC[‡], MIT[§], CMU[¶]

```
{bwilson, wqi, tagarwal, jlambert, jsingh, bpan, ratneshk, ahartnett,
   jpontes, dramanan, pcarr, jhays}@argo.ai, {skhandel}@cs.ubc.ca
```

## A    Supplementary Material

### A.1    Additional Information About Sensor Suite

In Figure 1, we provide a diagram of the sensor suite used to capture the Argoverse 2 datasets. Figure 2 shows the speed distribution for annotated pedestrian 3D cuboids and the yaw distribution.

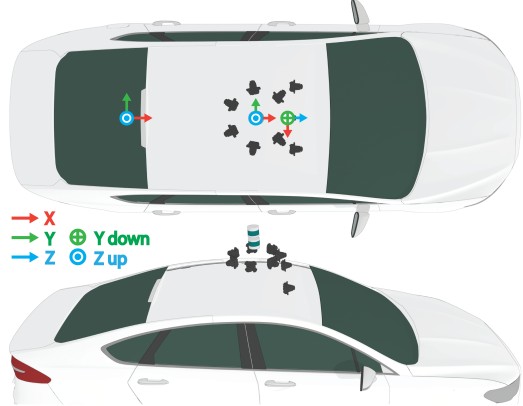

Figure 1: Car sensor schematic showing the three coordinate systems: (1) the vehicle frame in the rear axle; (2) the camera frame; and the lidar frame.

### A.2    Additional Information About Motion Forecasting Dataset

#### A.2.1    Interestingness Scores

Kinematic scoring selects for trajectories performing sharp turns or significant (de)accelerations. The map complexity program biases the data set towards trajectories complex traversals of the underlying lane graph. In particular, complex map regions, paths through intersections, and lane-changes score

---

[#]equal contribution

35th Conference on Neural Information Processing Systems (NeurIPS 2021) Track on Datasets and Benchmarks.

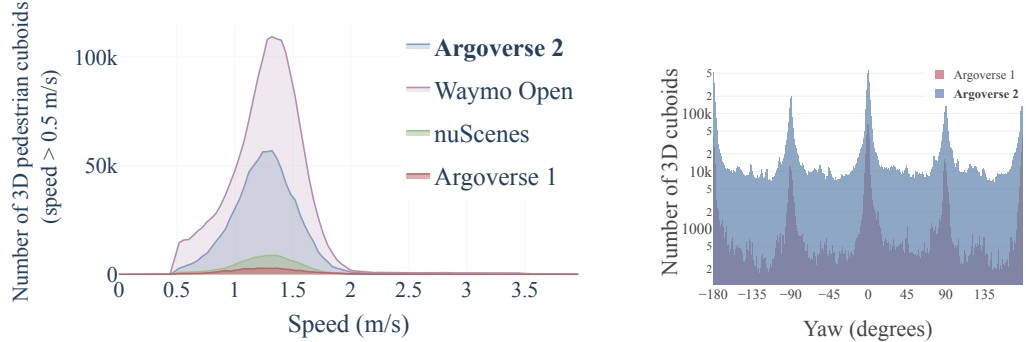

Figure 2: **Left**: Number of moving 3D cuboids for pedestrians by speed distribution. We define moving objects when the speed is greater than 0.5 m/s. **Right**: Number of annotated 3D cuboids by yaw distribution.

highly. Social scoring rewards tracks through dense regions of other actors. Social scoring also selects for non-vehicle object classes to ensure adequate samples from rare classes, such as motorcycles, for training and evaluation. Finally, the autonomous vehicle scoring program encourages the selection of tracks that intersect the ego-vehicle's desired route.

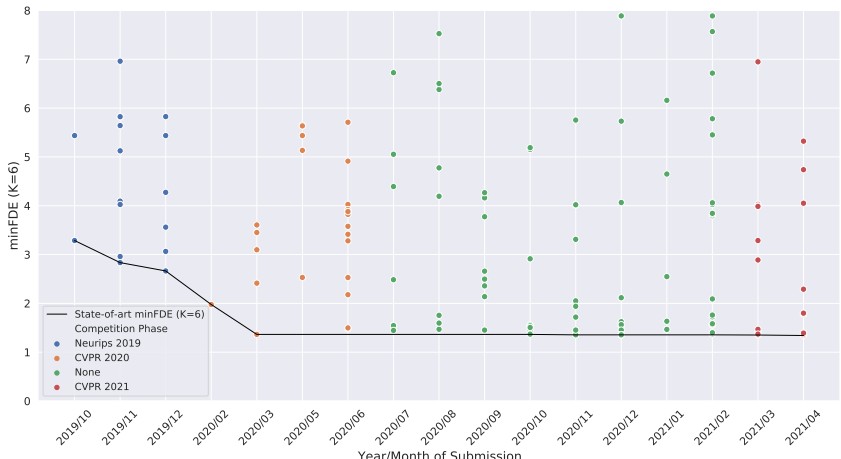

Figure 3: Metric values for submissions on Argoverse 1.1 over time. Individual points indicate submissions to the public leader board. Colors indicate specific competition phases. The solid black line indicates SOTA performance. The research community made massive gains which have plateaued since early 2020. However, we note that the number and diversity of methods performing at or near the SOTA continues to grow. Additionally, later competitions sorted the leaderboard by "Miss Rate" and probability weighted FDE, and those metrics showed progress. Still, minFDE did not improve.

## A.3   Additional Information About HD Maps

**Examples of HD maps from the Sensor Dataset**   In Figure 5, we display examples of local HD maps associated with individual logs/scenarios.

## A.4   Additional 3D Detection Results

In Figure 6, we show additional evaluation metrics for our detection baseline.

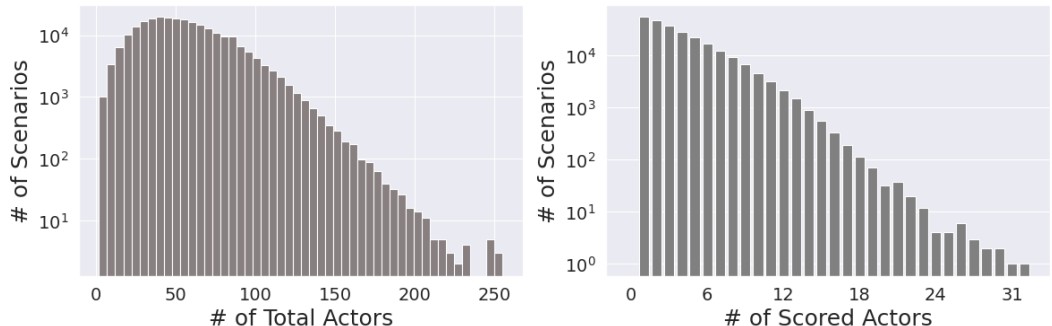

Figure 4: Histogram of the number of actors (both scored and all types) present in the Motion Forecasting Dataset scenarios. The Lidar Dataset is mined by the same criteria and thus follows the same distribution.

| MAP ENTITY | PROVIDED ATTRIBUTES | TYPE | DESCRIPTION |
|---|---|---|---|
| LANE SEGMENTS | IS_INTERSECTION | BOOLEAN | |
| | LANE TYPE | ENUMERATED TYPE | |
| | LEFT LANE BOUNDARY | 3D POLYLINE | THE POLYLINE OF THE LEFT BOUNDARY IN THE CITY MAP COORDINATE SYSTEM |
| | RIGHT LANE BOUNDARY | 3D POLYLINE | THE POLYLINE OF THE RIGHT BOUNDARY IN THE CITY MAP COORDINATE SYSTEM. |
| | LEFT LANE MARK TYPE | ENUMERATED TYPE | TYPE OF PAINTED LANE MARKING TO THE LEFT OF THE LANE SEGMENT ON THE ROAD. |
| | RIGHT LANE MARK TYPE | ENUMERATED TYPE | TYPE OF PAINTED LANE MARKING TO THE RIGHT OF THE LANE SEGMENT ON THE ROAD. |
| | LEFT NEIGHBOR | INTEGER | |
| | RIGHT NEIGHBOR | INTEGER | THE UNIQUE LANE SEGMENT IMMEDIATELY TO THE RIGHT OF SEGMENT, OR NONE. |
| | SUCCESSOR IDS | INTEGER LIST | LANE SEGMENTS THAT MAY BE ENTERED BY FOLLOWING FORWARD. |
| | TURN DIRECTION | ENUMERATED TYPE | NECESSARY TURN SIGNAL FOR A LANE SPLIT OR LEFT/ RIGHT TURN AT AN INTERSECTION |
| | ID | INTEGER | UNIQUE IDENTIFIER |
| DRIVABLE AREA | AREA BOUNDARY | 3D POLYGONS | AREA WHERE IT IS POSSIBLE FOR THE AV TO DRIVE WITHOUT DAMAGING ITSELF |
| | ID | INTEGER | UNIQUE IDENTIFIER |
| PEDESTRIAN CROSSINGS | EDGE1, EDGE2 | 3D POLYLINES | ENDPOINTS OF BOTH EDGE ALONG THE PRINCIPAL AXIS, THUS DEFINING A POLYGON. |
| | ID | INTEGER | UNIQUE IDENTIFIER |
| GROUND SURFACE HEIGHT | | 2D RASTER ARRAY | |

Table 1: HD map attributes for each Argoverse 2 scenario.

## A.5   Training Details of SPF2 baseline

We sample the 2-second training snippets (1 second past and 1 second future) every 0.5 second. Thus for a training log at the length of 30 seconds, there would be 59 training snippets being sampled. We train the model for 16 epochs by using the Adam optimizer with the learning rate of $4e-3$, betas of 0.9 and 0.999, and batch size of 16 per GPU.

## A.6   Additional Motion Forecasting Experiments

### A.6.1   Transfer Learning

The results of transfer learning experiments are summarized in Table 2. WIMP was trained and tested in different settings with Argoverse 1.1 and Argoverse 2. As expected, the model works best when it is trained and tested on the same distribution (i.e. both train and test data come from Argoverse 1.1, or both from 2). For example, when WIMP is tested on Argoverse 2 (6s), the model trained on Argoverse 2 (6s) has a *minFDE* of 2.91, whereas the one trained on Argoverse 1.1 (3s) has a *minFDE* of 6.82 (i.e. approx 2.3x worse). Likewise, in the reverse setting, when WIMP is tested on Argoverse 1.1 (3s), the model trained on Argoverse 1.1 (3s) has a *minFDE* of 1.14 and the one trained on Argoverse 2 (6s) has *minFDE* of 2.05 (i.e. approx 1.8x worse). This indicates that transfer learning from 2 to 1.1 is more useful than the reverse despite being smaller in the number of scenarios. Please note that it is a common practice to train and test sequential models on varied sequence length (eg. machine translation). As such, it is still reasonable to expect a model trained with 3s to do well on 6s horizon. Several factors may contribute to distribution shift, including differing prediction horizon, cities, mining protocols, object types. Notably, however, these results indicate that Argoverse 2 is significantly more challenging and diverse than its predecessor.

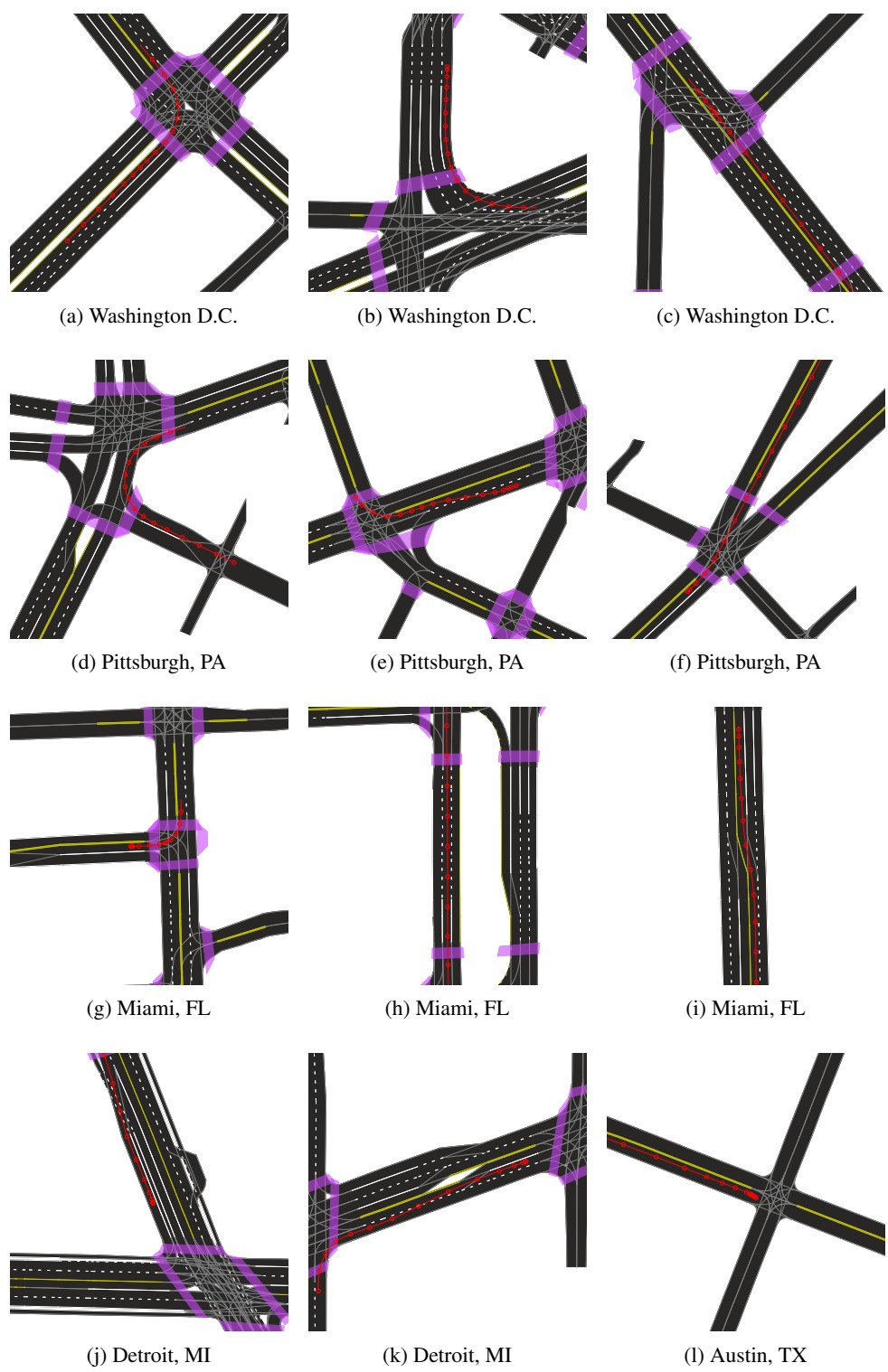

(a) Washington D.C.

(b) Washington D.C.

(c) Washington D.C.

(d) Pittsburgh, PA

(e) Pittsburgh, PA

(f) Pittsburgh, PA

(g) Miami, FL

(h) Miami, FL

(i) Miami, FL

(j) Detroit, MI

(k) Detroit, MI

(l) Austin, TX

Figure 5: Examples of egovehicle (AV) trajectories on local vector maps from the Sensor Dataset across several different cities. A 100m × 100m local map region is shown. Crosswalks are indicated in purple. Red circles denote the AV pose discretely sampled at 1 Hz for the purposes of illustration. Pose is provided at >20 Hz in the dataset, as indicated by the trajectory path indicated by a red line. City layouts vary dramatically, e.g. roadways in Miami are usually aligned parallel to a north-south, east-west grid, while roadways in Pittsburgh are generally not.

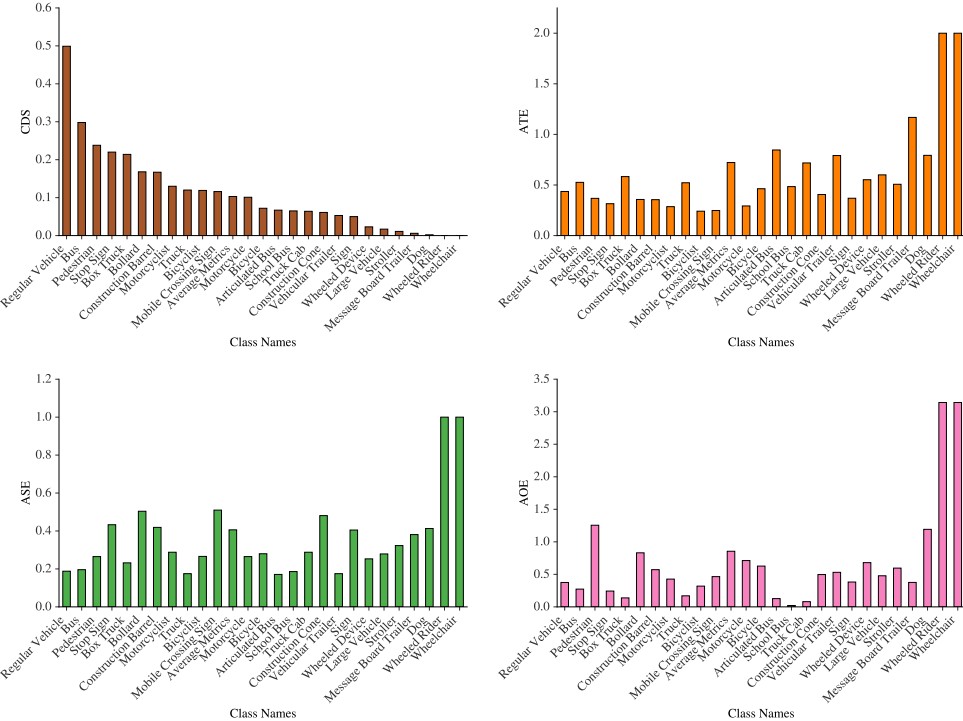

Figure 6: 3D object detection performance on the *validation* split of the *Sensor Dataset*. **Top Row:** Composite detection score (left). Average translation error (right) **Bottom Row:** Average scaling error (left), and average orientation error (right). Results are shown on the *validation* set of the *Sensor Dataset*.

### A.6.2   Experiment with different object types

Table 3 shows the results on Nearest Neighbor baseline (without map prior) on different object types. As one would expect, the displacement errors in pedestrians are significantly lower than other object types. This occurs because they move at significantly slower velocities. However, this does not imply that pedestrian motion forecasting is a solved problem and one should rather focus on other object types. This instead means that we need to come up with better metrics that can capture that fact lower displacement errors in pedestrians can often be more critical than higher errors in vehicles. We leave this line of work for future scope.

Table 2: Performance of WIMP when trained and tested on different versions of Argoverse. Training and evaluation is restricted to vehicle-like (vehicle, bus, motorcyclist) object types as only vehicles were present in Argoverse 1.1. All the results are for K=6, and prediction horizon is specified in parentheses. Notably, the model trained on a 3s horizon performs poorly on the longer 6s horizon.

| Train Split (pred. horizon) | Test Split (pred. horizon) | minADE ↓ | minFDE ↓ | MR ↓ |
|---|---|---|---|---|
| Argoverse 1.1 (3s) | Argoverse 1.1 (3s) | **0.75** | **1.14** | **0.12** |
| Argoverse 2 (6s) | Argoverse 1.1 (3s) | 1.68 | 2.05 | 0.27 |
| Argoverse 1.1 (3s) | Argoverse 2 (3s) | 0.94 | 1.88 | 0.26 |
| Argoverse 1.1 (3s) | Argoverse 2 (6s) | 4.93 | 6.82 | 0.77 |
| Argoverse 2 (6s) | Argoverse 2 (6s) | **1.48** | **2.91** | **0.43** |

## B   Datasheet for Argoverse 2

**For what purpose was the dataset created?**   Was there a specific task in mind? Was there a specific gap that needed to be filled? Please provide a description.
Argoverse was created to support the global research community in improving the state of the art in

Table 3: Performance of Nearest Neighbor baseline on different object types for K=6. The most accurately predicted object type for each evaluation metric is highlighted in bold.

| Object Type | #Samples | minADE ↓ | minFDE ↓ | MR ↓ |
|---|---|---|---|---|
| All | 9955 | 2.48 | 5.52 | 0.66 |
| Vehicle | 8713 | 2.62 | 5.87 | 0.70 |
| Bus | 439 | 2.69 | 5.59 | 0.73 |
| Pedestrians | 677 | **0.69** | **1.31** | **0.17** |
| Motorcyclist | 39 | 2.33 | 5.07 | 0.61 |
| Cyclist | 87 | 1.48 | 2.80 | 0.42 |

machine learning tasks vital for self driving. The Argoverse 2 datasets described in this manuscript improve upon the initial Argoverse datasets. These datasets support many tasks, from 3D perception to motion forecasting to HD map automation.

The three datasets proposed in this manuscript address different gaps in this space. See the comparison charts in the main manuscript for a more detailed breakdown.

The Argoverse 2 *Sensor Dataset* has a richer taxonomy than similar datasets. It is the only dataset of similar size to have stereo imagery. The 1,000 logs in the dataset were chosen to have a variety of object types with diverse interactions.

The Argoverse 2 *Motion Forecasting Dataset* also has a richer taxonomy than existing datasets. The scenarios in the dataset were mined with an emphasis on unusual behaviors that are difficult to predict.

The Argoverse 2 *Lidar Dataset* is the largest *Lidar Dataset*. Only the concurrent ONCE dataset is similarly sized to enable self-supervised learning in lidar space. Unlike ONCE, our dataset contains HD maps and high frame rate lidar.

**Who created this dataset (e.g., which team, research group) and on behalf of which entity (e.g., company, institution, organization)?**
The Argoverse 2 datasets were created by researchers at Argo AI.

**What support was needed to make this dataset?** (e.g.who funded the creation of the dataset? If there is an associated grant, provide the name of the grantor and the grant name and number, or if it was supported by a company or government agency, give those details.)
The creation of this dataset was funded by Argo AI.

**Any other comments?**
n/a

## COMPOSITION

**What do the instances that comprise the dataset represent (e.g., documents, photos, people, countries)?** Are there multiple types of instances (e.g., movies, users, and ratings; people and interactions between them; nodes and edges)? Please provide a description.

The three constituent datasets of Argoverse 2 have different attributes, but the core instances for each are brief "scenarios" or "logs" of 11 to 15 seconds that represent a continuous observation of a scene around a self-driving vehicle.

Each scenario in all three datasets has an HD map that includes lane boundaries, crosswalks, driveable area, etc. Scenarios for the *Sensor Dataset* additionally contain a raster map of ground height at .3 meter resolution.

**How many instances are there in total (of each type, if appropriate)?**
The *Sensor Dataset* has 1,000 15 second scenarios.

The *Lidar Dataset* has 20,000 15 second scenarios.

The *Motion Forecasting Dataset* has 100,000 11 second scenarios.

**Does the dataset contain all possible instances or is it a sample (not necessarily random) of instances from a larger set?** If the dataset is a sample, then what is the larger set? Is the sample representative of the larger set (e.g., geographic coverage)? If so, please describe how this representativeness was validated/verified. If it is not representative of the larger set, please describe why not (e.g., to cover a more diverse range of instances, because instances were withheld or unavailable).

The scenarios in the dataset are a sample of the set of observations made by a fleet of self-driving vehicles. The data is not uniformly sampled. The particular samples were chosen to be geographically diverse (spanning 6 cities - Pittsburgh, Detroit, Austin, Palo Alto, Miami, and Washington D.C.), to include interesting behavior (e.g. cars making unexpected maneuvers), to contain interesting weather (e.g. rain and snow), and to contain scenes with many objects of diverse types in motion (e.g. a crowd walking, riders on e-scooters splitting lanes between many vehicles, an excavator operating at a construction site, etc.).

**What data does each instance consist of?** "Raw" data (e.g., unprocessed text or images) or features? In either case, please provide a description.

Each *Sensor Dataset* scenario is 15 seconds in duration. Each scenario has 20 fps video from 7 ring cameras, 20 fps video from two forward facing stereo cameras, and 10 hz lidar returns from two out-of-phase 32 beam lidars. The ring cameras are synchronized to fire when either lidar sweeps through their field of view. Each scenario contains vehicle pose over time and calibration data to relate the various sensors.

Each *Lidar Dataset* scenario is 15 seconds in duration. These scenarios are similar to those of the *Sensor Dataset*, except that there is no imagery.

Each *Motion Forecasting* scenario is 11 seconds in duration. These scenarios contain no sensor data, but instead contain tracks of objects such as vehicles, pedestrians, and bicycles. The tracks specify the category of each object (e.g. bus or bicycle) as well as their location and heading at a 10 hz sampling interval.

The HD map associated with all three types of scenarios contains polylines describing lanes, crosswalks, and driveable area. Lanes form a graph with predecessors and successors, e.g. a lane that splits can have two successors. Lanes have precisely localized lane boundaries that include paint type (e.g. double solid yellow). Driveable area, also described by a polygon, is the area where it is possible but not necessarily legal to drive. It includes areas such as road shoulders.

**Is there a label or target associated with each instance?** If so, please provide a description.

Each *Sensor Dataset* scenario has 3D track annotations for dynamic objects such as vehicles, pedestrians, strollers, dogs, etc. The tracks are suitable as ground truth for tasks such as 3D object detection and 3D tracking. The 3D track labels are intentionally held out from the test set. The HD map could also be thought of as labels for each instance, and would be suitable as ground truth for lane detection or map automation. The vehicle pose data could be considered ground truth labels for visual odometry. The lidar depth estimates can act as ground truth for monocular or stereo depth estimation.

The *Lidar Dataset* does not have human annotations beyond the HD map. Still, the evolving point cloud itself can be considered ground truth for point cloud forecasting.

Each *Motion Forecasting Dataset* scenario has labels specifying the objects that are "agents". These objects have interesting behavior and are observed for the entirety of the scenario. These are the objects for which algorithms will be asked to forecast the future motion. The future motion of objects in each scenario is intentionally held out in the test set.

**Is any information missing from individual instances?** If so, please provide a description, explaining why this information is missing (e.g., because it was unavailable). This does not include intentionally removed information, but might include, e.g., redacted text.

In the *Sensor Dataset*, objects are only labeled within 5 meters of the driveable area. For example, a person sitting on their front porch will not be labeled.

In the *Sensor Dataset* and *Motion Forecasting Dataset*, instances are not necessarily labeled for the full duration of each scenario if the objects move out of observation range or become occluded.

**Z Are relationships between individual instances made explicit (e.g., users' movie ratings, social network links)?** If so, please describe how these relationships are made explicit.

The instances of the three datasets are disjoint. They each carry their own HD map for the region around the scenario. These HD maps may overlap spatially, though. For example, many forecasting scenarios may take place in the same intersection. If a user of the dataset wanted to recover the spatial relationship between scenarios, they could do so through the Argoverse API.

**Are there recommended data splits (e.g., training, development/validation, testing)?** If so, please provide a description of these splits, explaining the rationale behind them.

We define splits of each dataset. The *Sensor Dataset* is split 700 / 150 / 150 between train, validation, and test. The *Lidar Dataset* is split 16,000 / 2,000 / 2,000 and the *Motion Forecasting Dataset* is split 80,000 / 10,000 / 10,000. In all cases, the splits are designed to make the training dataset as large as possible while keeping the validation and test datasets large and diverse enough to accurately benchmark models learned on the training set.

**Are there any errors, sources of noise, or redundancies in the dataset?** If so, please provide a description.

Every sensor used in the dataset – ring cameras, stereo cameras, and lidar – has noise associated with it. Pixel intensities, lidar intensities, and lidar point 3D locations all have noise. Lidar points are also quantized to float16 which leads to roughly a centimeter of quantization error. 6 degree of freedom vehicle pose also has noise. The calibration specifying the relationship between sensors can be imperfect.

The HD map for each scenario can contain noise, both in terms of lane boundary locations and precise ground height.

The 3D object annotations in the *Sensor Dataset* do not always match the spatial extent and motion of an object in the real world. For example, we assume that objects do not change size during a scenario, but this could be violated by a car opening its door. 3D annotations for distant objects with relatively few pixels and lidar returns are less accurate.

The object tracks in the *Motion Forecasting* dataset are imperfect and contain errors typical of a real-time 3D tracking method. Our expectation is that a motion forecasting algorithm should operate well despite this noise.

**Is the dataset self-contained, or does it link to or otherwise rely on external resources (e.g., websites, tweets, other datasets)?** If it links to or relies on external resources, a) are there guarantees that they will exist, and remain constant, over time; b) are there official archival versions of the complete dataset (i.e., including the external resources as they existed at the time the dataset was created); c) are there any restrictions (e.g., licenses, fees) associated with any of the external resources that might apply to a future user? Please provide descriptions of all external resources and any restrictions associated with them, as well as links or other access points, as appropriate.

The data itself is self-hosted, like Argoverse 1 [see `https://www.argoverse.org/`], and we maintain public links to all previous versions of the dataset in case of updates. The data is independent of any previous datasets, including Argoverse 1.

**Does the dataset contain data that might be considered confidential (e.g., data that is protected by legal privilege or by doctor-patient confidentiality, data that includes the content of individuals' non-public communications)?** If so, please provide a description.

No.

**Does the dataset contain data that, if viewed directly, might be offensive, insulting, threatening, or might otherwise cause anxiety?** If so, please describe why.

No.

**Does the dataset relate to people?** If not, you may skip the remaining questions in this section.

Yes, the dataset contains images and behaviors of thousands of people on public streets.

**Does the dataset identify any subpopulations (e.g., by age, gender)?** If so, please describe how these subpopulations are identified and provide a description of their respective distributions within the dataset.
No.

**Is it possible to identify individuals (i.e., one or more natural persons), either directly or indirectly (i.e., in combination with other data) from the dataset?** If so, please describe how.
We do not believe so. All image data has been anonymized. Faces and license plates are obfuscated by replacing them with a 5x5 grid, where each grid cell is the average color of the original pixels in that grid cell. This anonymization is done manually and is not limited by our 3D annotation policy. For example, a person sitting on their front porch 10 meters from the road would not be labeled with a 3D cuboid, but their face would still be obscured.

**Does the dataset contain data that might be considered sensitive in any way (e.g., data that reveals racial or ethnic origins, sexual orientations, religious beliefs, political opinions or union memberships, or locations; financial or health data; biometric or genetic data; forms of government identification, such as social security numbers; criminal history)?** If so, please provide a description.
No.

**Any other comments?**
n/a

---

## COLLECTION

**How was the data associated with each instance acquired?** Was the data directly observable (e.g., raw text, movie ratings), reported by subjects (e.g., survey responses), or indirectly inferred/derived from other data (e.g., part-of-speech tags, model-based guesses for age or language)? If data was reported by subjects or indirectly inferred/derived from other data, was the data validated/verified? If so, please describe how.
The sensor data was directly acquired by a fleet of autonomous vehicles.

**Over what timeframe was the data collected?** Does this timeframe match the creation timeframe of the data associated with the instances (e.g., recent crawl of old news articles)? If not, please describe the timeframe in which the data associated with the instances was created. Finally, list when the dataset was first published.
The data was collected in 2020 and 2021. The dataset is not yet public, but will be public before NeurIPS 2021.

**What mechanisms or procedures were used to collect the data (e.g., hardware apparatus or sensor, manual human curation, software program, software API)?** How were these mechanisms or procedures validated?
The Argoverse 2 data comes from Argo 'Z1' fleet vehicles. These vehicles use Velodyne lidars and traditional RGB cameras. All sensors are calibrated by Argo. HD maps and 3D object annotations are created and validated through a combination of computational tools and human annotations. Object tracks in the *Motion Forecasting Dataset* are created by a 3D tracking algorithm.

**What was the resource cost of collecting the data?** (e.g. what were the required computational resources, and the associated financial costs, and energy consumption - estimate the carbon footprint. See Strubell *et al.* for approaches in this area.)
The data was captured during normal fleet operations, so there was minimal overhead for logging particular events. The transformation and post-processing of several terabytes of data consumed an estimated 1,000 machine hours. We estimate a Carbon footprint of roughly 1,000 lbs based on the CPU-centric workload.

**If the dataset is a sample from a larger set, what was the sampling strategy (e.g., deterministic, probabilistic with specific sampling probabilities)?**
The *Sensor Dataset* scenarios were chosen from a larger set through manual review. The *Lidar Dataset* and *Motion Forecasting Dataset* scenarios were chosen by heuristics which looked for interesting object behaviors during fleet operations.

**Who was involved in the data collection process (e.g., students, crowdworkers, contractors) and how were they compensated (e.g., how much were crowdworkers paid)?**
Argo employees and Argo interns curated the data. Data collection and data annotation was done by Argo employees. Crowdworkers were not used.

**Were any ethical review processes conducted (e.g., by an institutional review board)?** If so, please provide a description of these review processes, including the outcomes, as well as a link or other access point to any supporting documentation.
No.

**Does the dataset relate to people?** If not, you may skip the remainder of the questions in this section.
Yes.

**Did you collect the data from the individuals in question directly, or obtain it via third parties or other sources (e.g., websites)?**
The data is collected from vehicles on public roads, not from a third party.

**Were the individuals in question notified about the data collection?** If so, please describe (or show with screenshots or other information) how notice was provided, and provide a link or other access point to, or otherwise reproduce, the exact language of the notification itself.
No, but the data collection was not hidden. The Argo fleet vehicles are well marked and have obvious cameras and lidar sensors. The vehicles only capture data from public roads.

**Did the individuals in question consent to the collection and use of their data?** If so, please describe (or show with screenshots or other information) how consent was requested and provided, and provide a link or other access point to, or otherwise reproduce, the exact language to which the individuals consented.
No. People in the dataset were in public settings and their appearance has been anonymized. Drivers, pedestrians, and vulnerable road users are an intrinsic part of driving on public roads, so it is important that datasets contain people so that the community can develop more accurate perception systems.

**If consent was obtained, were the consenting individuals provided with a mechanism to revoke their consent in the future or for certain uses?** If so, please provide a description, as well as a link or other access point to the mechanism (if appropriate)
n/a

**Has an analysis of the potential impact of the dataset and its use on data subjects (e.g., a data protection impact analysis) been conducted?** If so, please provide a description of this analysis, including the outcomes, as well as a link or other access point to any supporting documentation.
No.

**Any other comments?**
n/a

---

## PREPROCESSING / CLEANING / LABELING

---

**Was any preprocessing/cleaning/labeling of the data done (e.g., discretization or bucketing, tokenization, part-of-speech tagging, SIFT feature extraction, removal of instances, processing of missing values)?** If so, please provide a description. If not, you may skip the remainder of the questions in this section.

Yes. Images are reduced from their full resolution. 3D point locations are quantized to float16. Ground height maps are quantized to .3 meter resolution from their full resolution. HD map polygon vertex locations are quantized to .01 meter resolution. 3D annotations are smoothed. For the *Motion Forecasting Dataset*, transient 3D tracks are suppressed and object locations are smoothed over time.

**Was the "raw" data saved in addition to the preprocessed/cleaned/labeled data (e.g., to support unanticipated future uses)?** If so, please provide a link or other access point to the "raw" data.

Yes, but such data is not public.

**Is the software used to preprocess/clean/label the instances available?** If so, please provide a link or other access point.

No.

**Any other comments?**

n/a

## USES

**Has the dataset been used for any tasks already?** If so, please provide a description.

Yes, this manuscript benchmarks a contemporary 3D object detection method on the *Sensor Dataset* and a contemporary motion forecasting method on the *Motion Forecasting Dataset*.

**Is there a repository that links to any or all papers or systems that use the dataset?** If so, please provide a link or other access point.

Not yet, because the dataset is not public. For the Argoverse 1 datasets, we maintain four leaderboards for 3D Tracking [https://eval.ai/web/challenges/challenge-page/453/overview], 3D Detection [https://eval.ai/web/challenges/challenge-page/725/overview], Motion Forecasting [https://eval.ai/web/challenges/challenge-page/454/overview], and Stereo Depth Estimation [https://eval.ai/web/challenges/challenge-page/917/overview]. Argoverse 1 was also used as the basis for a Streaming Perception challenge [https://eval.ai/web/challenges/challenge-page/800/overview]. We plan to add similar leaderboards for Argoverse 2.

**What (other) tasks could the dataset be used for?**

The datasets could be used for research on visual odometry, pose estimation, lane detection, map automation, self-supervised learning, structure-from-motion, scene flow, optical flow, time to contact estimation, pseudo-lidar, and point cloud forecasting.

**Is there anything about the composition of the dataset or the way it was collected and preprocessed/cleaned/labeled that might impact future uses?** For example, is there anything that a future user might need to know to avoid uses that could result in unfair treatment of individuals or groups (e.g., stereotyping, quality of service issues) or other undesirable harms (e.g., financial harms, legal risks) If so, please provide a description. Is there anything a future user could do to mitigate these undesirable harms?

No.

**Are there tasks for which the dataset should not be used?** If so, please provide a description.

The dataset should not be used for tasks which depend on faithful appearance of faces or license plates since that data has been obfuscated. For example, running a face detector to try and estimate

how often pedestrians use crosswalks will not result in meaningful data.

**Any other comments?**
n/a

---

## DISTRIBUTION

**Will the dataset be distributed to third parties outside of the entity (e.g., company, institution, organization) on behalf of which the dataset was created?** If so, please provide a description.
Yes, the dataset will be hosted on `https://www.argoverse.org/` like Argoverse 1 and 1.1.

**How will the dataset will be distributed (e.g., tarball on website, API, GitHub)?** Does the dataset have a digital object identifier (DOI)?
The dataset will be distributed as a series of tar.gz files as was the case for Argoverse 1 and Argoverse 1.1. See `https://www.argoverse.org/data.html#download-link`. The files are broken up to make the process more robust to interruption (e.g. a single 2 TB file failing after 3 days would be frustrating) and to allow easier file manipulation (an end user might not have 2 TB free on a single drive, and if they do they might not be able to decompress the entire file at once).

**When will the dataset be distributed?**
The data will be available for download before NeurIPS 2021.

**Will the dataset be distributed under a copyright or other intellectual property (IP) license, and/or under applicable terms of use (ToU)?** If so, please describe this license and/or ToU, and provide a link or other access point to, or otherwise reproduce, any relevant licensing terms or ToU, as well as any fees associated with these restrictions.
Yes, the dataset will be released under the same Creative Commons license as Argoverse 1 – CC BY-NC-SA 4.0. Details can be seen at `https://www.argoverse.org/about.html#terms-of-use`.

**Have any third parties imposed IP-based or other restrictions on the data associated with the instances?** If so, please describe these restrictions, and provide a link or other access point to, or otherwise reproduce, any relevant licensing terms, as well as any fees associated with these restrictions.
No.

**Do any export controls or other regulatory restrictions apply to the dataset or to individual instances?** If so, please describe these restrictions, and provide a link or other access point to, or otherwise reproduce, any supporting documentation.
No.

**Any other comments?**
n/a

---

## MAINTENANCE

**Who is supporting/hosting/maintaining the dataset?**
Argo AI

**How can the owner/curator/manager of the dataset be contacted (e.g., email address)?**
The Argoverse team responds through the Github page for the Argoverse API: `https://github.com/argoai/argoverse-api/issues`. It currently contains 2 open issues and 126 closed issues.

For privacy concerns, contact information can be found here: `https://www.argoverse.org/about.html#privacy`

**Is there an erratum?** If so, please provide a link or other access point.
No.

**Will the dataset be updated (e.g., to correct labeling errors, add new instances, delete instances)?** If so, please describe how often, by whom, and how updates will be communicated to users (e.g., mailing list, GitHub)?
It is possible that the constituent Argoverse 2 datasets are updated to correct errors. This was the case with Argoverse 1 which was incremented to Argoverse 1.1. Updates will be communicated on Github and through our mailing list.

**If the dataset relates to people, are there applicable limits on the retention of the data associated with the instances (e.g., were individuals in question told that their data would be retained for a fixed period of time and then deleted)?** If so, please describe these limits and explain how they will be enforced.
No.

**Will older versions of the dataset continue to be supported/hosted/maintained?** If so, please describe how. If not, please describe how its obsolescence will be communicated to users.
Yes. We still host Argoverse 1 even though we have declared it "deprecated". See `https://www.argoverse.org/data.html#download-link`. We will use the same warning if we ever deprecate Argoverse 2. Note: Argoverse 2 does not deprecate Argoverse 1. They are independent collections of datasets.

**If others want to extend/augment/build on/contribute to the dataset, is there a mechanism for them to do so?** If so, please provide a description. Will these contributions be validated/verified? If so, please describe how. If not, why not? Is there a process for communicating/distributing these contributions to other users? If so, please provide a description.
Yes. For example, the streaming perception challenge was built by CMU researchers who added new 2D object annotations to Argoverse 1.1 data. The Creative Commons license we use for Argoverse 2 ensures that the community can do the same thing without needing Argo's permission.

We do not have a mechanism for these contributions/additions to be incorporated back into the 'base' Argoverse 2. Our preference would generally be to keep the 'base' dataset as is, and to give credit to noteworthy additions by linking to them as we have done in the case of the Streaming Perception Challenge (see link at the top of this Argoverse page `https://www.argoverse.org/tasks.html`).

**Any other comments?**
n/a

## C    Additional Supplementary Requirements

1. Submission introducing new datasets must include the following in the supplementary materials:

    (a) Dataset documentation and intended uses. Recommended documentation frameworks include datasheets for datasets, dataset nutrition labels, data statements for NLP, and accountability frameworks.
    **See Datasheet above.**

    (b) URL to website/platform where the dataset/benchmark can be viewed and downloaded by the reviewers.
    **Private URL is provided to reviewers. Dataset will be made public on Argoverse.org before camera ready.**

(c) Author statement that they bear all responsibility in case of violation of rights, etc., and confirmation of the data license.
**Authors confirm that they are responsible for the contents of the dataset and are responsible for any violation of rights. Authors confirm that the dataset will be released under Creative Commons license CC BY-NC-SA 4.0**

(d) Hosting, licensing, and maintenance plan. The choice of hosting platform is yours, as long as you ensure access to the data (possibly through a curated interface) and will provide the necessary maintenance. **The dataset will be hosted on Argoverse.org, as Argoverse version 1 is. Our datasets require no user registration for access. The license is CC BY-NC-SA 4.0. See Datasheet for more details.**

2. To ensure accessibility, the supplementary materials for datasets must include the following:

(a) Links to access the dataset and its metadata. This can be hidden upon submission if the dataset is not yet publicly available but must be added in the camera-ready version. In select cases, e.g when the data can only be released at a later date, this can be added afterward. Simulation environments should link to (open source) code repositories.
**Private URL is provided to reviewers. Dataset will be made public on Argoverse.org before camera ready.**

(b) The dataset itself should ideally use an open and widely used data format. Provide a detailed explanation on how the dataset can be read. For simulation environments, use existing frameworks or explain how they can be used.
**The dataset can be read with the Argoverse API. See** `https://github.com/argoai/argoverse-api` **for details on usage.**

(c) Long-term preservation: It must be clear that the dataset will be available for a long time, either by uploading to a data repository or by explaining how the authors themselves will ensure this.
**In addition to long term hosting on Argoverse.org, the Creative Commons license enables rehosting by any repository. The authors will ensure that the dataset is accessible.**

(d) Explicit license: Authors must choose a license, ideally a CC license for datasets, or an open source license for code (e.g. RL environments).
**The dataset will be released under Creative Commons license CC BY-NC-SA 4.0**

(e) Add structured metadata to a dataset's meta-data page using Web standards (like schema.org and DCAT): This allows it to be discovered and organized by anyone. If you use an existing data repository, this is often done automatically.
**The dataset's metadata page will include structured metadata.**

(f) Highly recommended: a persistent dereferenceable identifier (e.g. a DOI minted by a data repository or a prefix on identifiers.org) for datasets, or a code repository (e.g. GitHub, GitLab,...) for code. If this is not possible or useful, please explain why.
**Argoverse API code is hosted on Github.**

**Environmental Impact Statement.** Amount of Compute Used: We estimate 2,000 CPU and 500 GPU hours were used in the collection of the dataset and the performance of baseline experiments.