# OpenReview forum: "Argoverse 2: Next Generation Datasets for Self-Driving Perception and Forecasting"
_NeurIPS.cc/2021/Track/Datasets_and_Benchmarks/Round2 — NeurIPS 2021 Datasets and Benchmarks Track (Round 2)_

### Official Review · Reviewer_dTxt · 2021-09-10
**A novel self-driving dataset yet more discussions are required.**

**Rating:** 7
**Confidence:** 4
**Correctness:** Yes. It looks good in terms of correc…
**Clarity:** Yes. The paper is well written yet th…

**Strengths:**

1. The three datasets are more diverse than the existing datasets in terms of cities, categories, and scenarios. The authors include more cities to deal with the domain gap between different cities. Meanwhile, the authors introduce the largest taxonomy to date for sensor and forecasting datasets, and they ensure enough samples of rare objects to train and evaluate models. In addition, the authors mine difficult scenarios for the perception and forecasting tasks.

2. All the datasets are equipped with HD maps with 3D lane geometry, paint markings, crosswalks, higher resolution ground height, etc.

3. The authors control the size of the sensor dataset for the research community.

4. The experiments on point cloud forecasting and motion prediction are convicing.

**Weaknesses:**

1. The main weakness of this work is lack of in-depth analysis of the perception dataset. The authors claim that their datasets are for perception and forecasting, however, the experimental results are not reported on the perception tasks. Therefore, it is hard to quantitatively evaluate the value of the Argoverse 2.0 regarding the perception. For example, can the state-of-the-art detectors and trackers still perform satisfactorily in the especially crowded, dynamic, and kinematically unusual scenarios? Or can they generalize well across different cities?  Or can they achieve good performance when dealing with the new categories? It will be nice if the authors can provide more evidence regarding the perception dataset because the authors claim that Argoverse 2.0 is the next generation datasets for both perception and forecasting, but it seems that the experiments and discussions are only focused on the latter. If it is non-trivial to include comprehensive experiments regarding the perception tasks, the authors may add a justification for that in the text or improve the organization by offering more discussions around the perception.

2. The second weakness of this work is that the authors provide three different datasets for different purposes for the community. Actually, there are a lot of datasets in the current community, for example, nuScenes for multi-sensor perception, Waymo for large-scale perception, ONCE for self-supervised learning, etc. I believe that as a community, we are in need of a dataset that forms a union of strengths of existing datasets to innovate multi-sensor multi-task self-driving systems. However, Argoverse 2.0 further introduces three separate datasets. Can the authors merge the sensor and forecasting datasets or why do you build two separate datasets for perception and prediction although there are many works on joint perception and prediction tasks?  Although I understand that it is non-trivial to collect and annotate a large-scale multi-task dataset, I think the first and third one could be combined to facilitate joint perception and prediction research. More importantly, I always admire simple and elegant solutions instead of ones which improve the complexity for the community.

3. The third weakness is that the limitations are not well illustrated. How can we make reasonable assumptions about the object extent without accurate annotations given that different cities may have objects with different sizes? Why do we need to estimate the 3D bounding boxes in the motion forecasting dataset? How does the imperfect tracking influence the dataset? Do you use off-the-shelf trackers for annotations? How to deal with this problem? I think more illustrations are required regarding the dataset limitations, otherwise the users will be skeptical about the new datasets.

-------
In summary, I appreciate the hard work of the authors for this nice dataset. However, I will be more confident if the authors can offer more results or discussions regarding the perception tasks, clarify the reason why they build three separate datasets, and illustrate the dataset limitations comprehensively.

**Additional Feedback:**

I appreciate the hard work of the authors for this nice dataset. However, I will be more confident if the authors can offer more results or discussions regarding the perception tasks, clarify the reason why they build three separate datasets, and illustrate the dataset limitations comprehensively. I will raise my score once the authors address my concerns.
***
The authors addressed my concerns, and they also improved the paper with more results and discussions. So I would like to raise my score from 5 to 7.

**Documentation:**

Yes. The website looks comprehensive and professional.

**Ethics:**

I do not see any ethical concerns for this dataset. All faces and license plates, whether inside vehicles or outside of the driveable area, are blurred extensively to preserve privacy.

**Relation To Prior Work:**

Yes. The relation to prior works are clear yet some claims are not supported.

**Summary And Contributions:**

This work proposed three nice self-driving datasets: a multimodal sensor dataset for perception tasks, a large-scale Lidar dataset for self-supervised learning tasks, and a challenging motion forecasting dataset for prediction tasks. All three datasets are equipped with HD Maps. Overall, this work can be useful for the self-driving community since the proposed datasets are more diverse in terms of categories, cities, and scenarios compared to existing datasets. However, more analyses around the dataset are needed and more experiments should be conducted to verify the distinct advantage of the dataset.

---

> ### Author Response · Authors · 2021-09-30
> **Response to R4**
>
> Thank you for the detailed feedback. We have tried to address the shortcomings that you identified.
>
> ### Q1: ...experimental results are not reported on the perception tasks... Or can they achieve good performance when dealing with the new categories
>
> Thanks for the advice. See our shared response about detection baseline experiments.
>
> ### Q2: Can the authors merge the sensor and forecasting datasets or why do you build two separate datasets for perception and prediction although there are many works on joint perception and prediction tasks?
> This is a good point. It would be nice to have a single dataset with the combined attributes of the three datasets we propose. Data logistics have stopped us from doing this so far -- A sensor dataset scenario is an order of magnitude larger than a lidar dataset scenario, which is in turn two orders of magnitude larger than a forecasting dataset scenario.
>
> For motion forecasting, we see benefits in training on hundreds of thousands of scenarios, so we wouldn't want to shrink it to the size of the other datasets. In the other direction, we can't feasibly release or experiment with 100k sensor scenarios. The 1k sensor dataset and 20k lidar dataset are already a couple of Terabytes each.
>
> Certainly, there is nothing stopping someone from benchmarking motion forecasting on a sensor or lidar dataset. The sensor data has trivial ground truth, in the form of human object annotations, and based on your suggestion we will try to release a "ground truth", derived from tracking, for the lidar dataset, as well.
>
> ### Q3 (A): How can we make reasonable assumptions about the object extent without accurate annotations given that different cities may have objects with different sizes?
> While this has been shown to be an issue between countries (see [Train in Germany, Test in the USA et al, CVPR ‘20](https://openaccess.thecvf.com/content_CVPR_2020/html/Wang_Train_in_Germany_Test_in_the_USA_Making_3D_Object_CVPR_2020_paper.html)), our Sensor Dataset is collected from 6 cities within a single country (the United States).
>
> In the table below, we provide statistics from the Sensor Dataset about the dimensions of vehicles (our "REGULAR_VEHICLE" category) in each of the 6 cities. The numbers indicate that for vehicles, our assumptions about common object extent across many cities in the United States are reasonable.
>
> | City | Avg. length | Avg. width | Avg. height |
> | :--:| :--: | :--: | :--: |
> | PIT | 4.41 | 1.93 | 1.75 |
> | MIA | 4.45 | 1.94 | 1.69 |
> | DTW | 4.51 | 1.94 | 1.74 |
> | PAO | 4.50 | 1.96 | 1.71 |
> | WDC | 4.48 | 1.93 | 1.74 |
> | PAO | 4.50 | 1.96 | 1.71 |
> | ATX | 4.50 | 1.94 | 1.75 |
>
>
> ### Q3 (B): Why do we need to estimate the 3D bounding boxes in the motion forecasting dataset?
> We do not necessarily need to estimate 3D bounding boxes to have a meaningful motion forecasting task.  Argoverse 1.1 relied on point estimates only.  However, we thought it was important to include stereotyped bounding boxes in Argoverse 2.0 for multiple reasons.  First, several BEV rasterization based methods are difficult to test or deploy without such boxes.  In fact we saw multiple submissions to Argoverse 1.1 competitions which heavily preprosseed the data to add these bounding boxes.  Second, for many interactions (such as entering a lane), knowing even approximate bumper positions provides better signal than center of mass coordinates.  Finally, bounding boxes are a useful tool for visualization and understanding the data.  We think the usability of the dataset is extremely important and we found that bounding boxes contributed significantly to the visual understanding of the scene.  Please see the additional videos we have added to the private url.
>
> ### Q3 (C): How does the imperfect tracking influence the dataset?
> Our intent was for the forecasting task to accurately represent the on car problem.  Due to sensor range, occlusion, etc. tracks derived from on car observations are necessarily imperfect.  Impactful approaches will include robustness to these imperfections.  There is of course a baseline level of data quality required for any meaningful autonomous capabilities.  We invite reviewers to view videos of selected scenarios on the private webpage.  You will see sporadic heading errors or even an occasional object classification error.  However, track quality is generally quite high with consistent ids and smooth motion.
>
>
> ### Q3 (D): Do you use off-the-shelf trackers for annotations? How to deal with this problem?
> We use an auto-regressive tracker with no lookahead that we believe is representative of current real-time methods.

---

> > ### Author Response · Authors · 2021-09-30
> > **Response to R4**
> >
> > ### Q3 (E): I think more illustrations are required regarding the dataset limitations, otherwise the users will be skeptical about the new datasets.
> > We would like to address this concern. Would you like to see a characterization of the tracking noise in the main manuscript? By “illustrations” do you mean qualitative examples? We added video previews of motion forecasting scenarios to the private reviewer URL which do exhibit tracking noise. The tracking errors are typical of real time trackers -- tracks are lost for occluded objects, velocity estimates have noise, initial heading estimates for a newly seen object can be wrong but typically correct over time, and objects can be misclassified. All of these errors are more pronounced for objects further away from the AV.

---

### Official Review · Reviewer_Bgwn · 2021-09-15
**Argoverse 2.0 is a solid dataset but lacks some highlights compared to other self-driving datasets**

**Rating:** 6
**Confidence:** 5
**Correctness:** The dataset is constructed in a sound…
**Clarity:** The paper is well written.

**Strengths:**

Argoverse 2.0 have some strengths compared to previous datasets:

(1) The dataset contains crowded, dynamic, and kinematically unusual scenarios.

(2) The dataset contains 32 categories. The number of categories is much more than Waymo (4) and nuScenes (23).

(3) The dataset contains HD maps for each scene.

(4) The dataset contains large-scale lidar data for self-supervised learning.

**Weaknesses:**

(1) A major concern about Argoverse 2.0 is that it is quite similar to Waymo and nuScenes. Both nuScenes and Waymo have motion forecasting tasks. NuScenes also selected special scenarios and contains HD maps similar to Argoverse 2.0, and its 23 categories can cover most types of objects on the roads. Waymo has a similar scale of data compared to Argoverse 2.0, with 104k scenarios and 7.64M tracks compared to 100k scenarios and 7.27M tracks of Argoverse 2.0. Waymo also has more driving hours (574h vs. 305h) than Argoverse 2.0. Thus it seems that Argoverse 2.0 has no significant advantages on the data scale and modalities over previous self-driving datasets.

(2) The authors claim that their data is more diverse than other datasets, as they designed a special mechanism to only select those interesting scenarios into their dataset. However, there is no direct diversity comparison, neither qualitatively nor quantitatively, with other datasets. Thus it is unclear that to what extent Argoverse 2.0 is more diverse than Waymo and nuScenes, considering the fact that nuScenes also contains specially selected scenarios.

(3) Argoverse 2.0 has a lidar dataset for self-supervised learning, which is similar to the recently proposed ONCE dataset. Argoverse 2.0 has 6M point clouds at 10Hz and ONCE has 1M point clouds at 2Hz. Considering the sampling rate, their data scales in time are actually quite similar. ONCE provides a comprehensive benchmark of multiple self-supervised learning methods, while Argoverse 2.0 didn't provide any baseline results on the lidar dataset. I think baseline results on the lidar dataset are very important and should be provided.

**Additional Feedback:**

Here are some suggestions:

(1) Diversity comparison. The authors can give some concrete examples to illustrate how their dataset is more diverse than previous datasets, or design some diversity comparison experiments.

(2) The authors should at least provide some self-supervised learning results on their lidar dataset. It seems that only point cloud and motion forecasting results are presented.

**Documentation:**

The dataset contains sufficient details on data collection and organization, availability and maintenance, and ethical and responsible use.

**Ethics:**

No ethical concerns.

**Relation To Prior Work:**

The Argoverse 2.0 motion forecasting dataset is similar to the Waymo and nuScenes datasets. The Argoverse 2.0 lidar dataset is similar to the ONCE dataset. The advantages of using Argoverse 2.0 instead of those previous datasets are not clearly presented.

**Summary And Contributions:**

The paper presents a self-driving dataset for perception and forecasting. The dataset contains two parts: a Lidar dataset and a motion forecasting dataset.

---

> ### Author Response · Authors · 2021-09-30
> **Response to R3**
>
> Thank you for the feedback. We have tried to address your concerns in our updated manuscript. We respond to individual concerns below.
>
> ### Q1: Thus it seems that Argoverse 2.0 has no significant advantages on the data scale and modalities over previous self-driving datasets
>
> We intentionally do not try to expand the scale of the data (see paper line 36-40). Still, there are many distinguishing features (See lines 112-118). Our sensor dataset has a unique modality - stereo. Our sensor dataset has 10 times as many annotated cuboids as nuScenes (5x of that is explained by higher time resolution annotations), and 2x as many categories with suitable number of cuboids for training and evaluation. Our sensor dataset lidar is twice as dense and longer range, as well. Our evaluation range (100m vs 50m) and annotation range extend beyond nuScenes, too. All that said, we're definitely not arguing that our sensor dataset is superior to nuScenes. nuScenes is hugely influential (850+ citations already) and we think there is room for other complementary sensor datasets.
>
>
> ### Q2:  Diversity comparison - Unclear that to what extent Argoverse 2.0 is more diverse than Waymo and nuScenes, considering the fact that nuScenes also contains specially selected scenarios.
>
> See shared response about dataset comparisons.
>
>
> ### Q3: ONCE provides a comprehensive benchmark of multiple self-supervised learning methods, while Argoverse 2.0 didn't provide any baseline results on the lidar dataset
>
> We do have baseline self-supervised learning results for the point cloud forecasting task which shows the benefit of large scale training. This task also relies on the full framerate lidar that our dataset contains, and ONCE does not. ONCE is an impressive dataset, but keep in mind that it is concurrent work released this summer and submitted to this same NeurIPS dataset track.
>
> ### Additional Feedback #2: The authors should at least provide some self-supervised learning results on their lidar dataset. It seems that only point cloud and motion forecasting results are presented
>
> Thanks for the advice. Again, we wish to emphasize that point cloud forecasting IS a self supervised learning task, but we will try to add another for the final manuscript.

---

> > ### Comment · Reviewer_Bgwn · 2021-09-30
> > **Response of R3**
> >
> > **Dataset Comparisons (Q1)**
> >
> > What I was asking in W1 is mainly on the motion forecasting dataset. As I mentioned in W1, Waymo has a similar scale of data compared to Argoverse 2.0, with 104k scenarios and 7.64M tracks compared to 100k scenarios and 7.27M tracks of Argoverse 2.0. Waymo also has more driving hours (574h vs. 305h) than Argoverse 2.0.
> >
> > **Self-Supervised Learning**
> >
> > Since you have provided a large amount of raw data, it's good to see some baseline results that can leverage the large-scale unlabeled data to boost the performance of downstream perception and prediction tasks. I understand point cloud forecasting can be learned in a self-supervised manner, but this may restrict the application scope of the lidar dataset, as it only supports those tasks that can be learned in a self-supervised manner, and those tasks are not that many.

---

> > > ### Author Response · Authors · 2021-09-30
> > > **Response to R3**
> > >
> > > Thanks for responding! Waymo's motion dataset is impressive and we hope our motion forecasting dataset is complementary. Below we reproduced our response to R1 about motion dataset comparisons. We do plan to grow our motion forecasting dataset before release. Our compact representation will allow us to have more scenarios without the dataset becoming unwieldy.
> > >
> > > ***Comparison to Waymo***: Argoverse 2.0 and Waymo are complementary datasets. There is significant value in having two similar high-quality datasets that come from totally non-overlapping distributions. That said, Argoverse has a much more accessible entry point - our ~30GB vs their ~1.3TB.
> > >
> > > ***Comparison to Argoverse 1.1***: This dataset was the first motion-forecasting specific dataset in the self-driving domain and was pivotal in influencing increased research activity in this domain. However, as evident in Figure 2 of supplementary material, the performance of forecasting methods has saturated, with no significant improvement in minFDE over the last several months. Furthermore, the lack of object categories and multi-agent evaluation, as well as shorter forecast horizon, and smaller quantity of challenging scenarios, has limited its use. Argoverse 2.0 overcomes all these shortcomings and provides a much more “complete” dataset to work with.
> > >
> > > ***Comparison to Lyft***: Despite the fact that the Lyft dataset is the largest (measured by total log hours) on paper, the entire dataset was collected from a single 10 km stretch of road.
> > >
> > > ***Comparison to nuScenes***: The nuScenes prediction dataset tackles a slightly different flavor of motion forecasting compared to Argoverse and Waymo. Observations are at a much lower frequency than almost every other dataset (2 Hz vs. 10 Hz). This means that some scenarios have very few observations (potentially as few as 1), making the prediction task quite difficult. This difficulty is reflected in the best ADE numbers reported on their leaderboard, which are significantly higher than both Argoverse and Waymo. Further, their prediction dataset is mainly a repurposed version of tracking dataset, as such it spans 5.5h compared to 305h of Argoverse 2.0. This means that there are going to be a large number of overlapping scenarios and not enough “new” observations to learn from.
> > >
> > > **Self-supervised learning in the lidar domain**
> > > We think that self-supervision on the Lidar Dataset will transfer to more familiar lidar processing tasks like detection and segmentation. We also think more "pretext" tasks will emerge now that large lidar datasets are available, both from Argoverse and ONCE. The development of self-supervised learning in the image and text domains was catalyzed by the availability of nearly endless amounts of data from the internet. But there is no equivalent fountain of data for lidar, so these datasets are important. For pretext tasks based on dynamic evolution of scenes, we think our 10 hz sampling rate is valuable compared to ONCE's 2 hz sampling rate. Another form of self-supervision in our lidar dataset could come from our HD maps, which identify all road points, provide height about the ground for other points, and identify lane marking types and crosswalk locations. ONCE does not contain HD maps. We are investigating more self-supervised learning and transfer learning tasks with this dataset and will include results in the manuscript, time permitting.

---

### Official Review · Reviewer_b7Zs · 2021-09-17
**Review for ID 273**

**Rating:** 7
**Confidence:** 4
**Correctness:** The overall structure of this benchma…

**Strengths:**

+ Compared to Argoverse 1.1, this dataset shows a significant extension in terms of diversity, density, and quality. It can be used to develop a prediction model for vehicle and pedestrian detection, localization, motion forecasting, and 3D scene reconstruction from a challenging scenario.

+ The dataset is built upon a solid guiding principle which is considering practicality and diversity. This guiding principle addresses the limitations of existing datasets and provides guidance for future data collection.


**Weaknesses:**

- Weak justification for the necessity of the proposed dataset over existing ones. As described in Table 1, there exist many datasets for autonomous driving which seems already sufficient to develop a prediction model. It is not clear why and how the proposed dataset can enable a new function or better at improving the prediction accuracy than the dataset collected from all previous works. Simply showing the diversity from the histogram (Figure 2 or Figure 3) or table (Table 2) does not mean the dataset is indeed more useful to improve the model accuracy than the whole previous dataset. An effective cross-dataset evaluation can be a solution to demonstrate them.

- Weak motivation for 3D point cloud forecasting task. As a proof-of-concept for the validity of the unlabeled data, this paper introduces a 3D point cloud forecasting task. However, it is not fully motivated why this task is important in the autonomous driving scene. Why do we have to predict a complete 3D geometry of a vehicle for motion forecasting in a driving scene?

- Weak quantitative and qualitative demonstration of how challenging this dataset is. As described in L042, this paper claims that it includes more challenging data. However, it is difficult to understand why this dataset includes more challenging scenarios in what sense. Particularly, the insufficient visual demonstration that highlights such a level of difficulty is disappointing.

- Lack of experiment. The 3D bounding box dataset is not validated through experiments. For example, as similar to Argoverse 1.1, the authors can collect various baseline methods and perform train/test on this new dataset. This will provide an important starting point for the research in the 3D bounding box detection domain.


**Additional Feedback:**

Please address the comments in the weakness section.

**Clarity:**

Overall, the paper is well written and easy to follow.


**Documentation:**

This paper includes sufficient detail on data collection, organization, availability, and maintenance.

**Relation To Prior Work:**

Missing literature related to the visual perception for autonomous driving:
[1] Choi et al. "All-day visual place recognition: Benchmark dataset and baselines" CVPRW 2015
[2] Choi et al. "KAIST multi-spectral day/night data set for autonomous and assisted driving" T-ITS


**Summary And Contributions:**

This paper introduces a large collection of datasets for visual perception and motion forecasting in autonomous driving scenarios. It introduces the three chunks of dataset captured from six distinct cities: 1) Sensor dataset: 1,000 sequences of multimodal data which includes HD images from 9 cameras, lidar data, pose map, and associated annotation. 2) Lidar dataset: 20,000 sequences of unlabeled lidar data, and pose map. 3) Forecasting dataset: 100,000 scenarios of local map and annotation which include the interaction between the autonomous vehicle and other actors. The usability of the proposed dataset is validated by designing a deep learning model for the point cloud and motion forecasting prediction.

---

> ### Author Response · Authors · 2021-09-30
> **Response to R2**
>
> Thanks for the advice which has made the paper stronger.
>
> ### Q2A: this paper introduces a 3D point cloud forecasting task
>
> We actually didn’t introduce the 3D point cloud forecasting task. Several papers already explore this (See line 242 of the main text and [Weng et al., [CoRL ‘20]](https://arxiv.org/abs/2003.08376). Point-cloud forecasting is a self-supervised learning with growing community interest. We’ve updated the text to clarify this.
>
> ### Q2B: Why do we have to predict a complete 3D geometry of a vehicle for motion forecasting in a driving scene?
>
> We aren’t claiming that self-supervised point cloud forecasting is the future of motion forecasting and motion planning. But it may be that a “direct” approach, which skips the detection and tracking steps in typical forecasting problem definitions, is worthwhile in some situations. There have been other recent works that also try more “direct” approaches to predicting AV scene evolution, such as [Badki et al. [CVPR ‘21]](https://openaccess.thecvf.com/content/CVPR2021/html/Badki_Binary_TTC_A_Temporal_Geofence_for_Autonomous_Navigation_CVPR_2021_paper.html) (Best Student Paper Honorable Mentions at CVPR 2021) and [Hu et al. [CVPR ‘21]](https://openaccess.thecvf.com/content/CVPR2021/html/Hu_Safe_Local_Motion_Planning_With_Self-Supervised_Freespace_Forecasting_CVPR_2021_paper.html). It may also be the case that models learned for self-supervised point cloud forecasting transfer well to other lidar scene understanding tasks such as detection and semantic segmentation.
>
> ### Q3: Weak quantitative and qualitative demonstration of how challenging this dataset is. As described in L042, this paper claims that it includes more challenging data. However, it is difficult to understand why this dataset includes more challenging scenarios in what sense. Particularly, the insufficient visual demonstration that highlights such a level of difficulty is disappointing.
> See shared response about dataset comparisons. Due to your concern about visualizations, we have added video visualizations of motion forecasting scenarios to the private review page.
>
> ### Q4: Lack of experiment. The 3D bounding box dataset is not validated through experiments.
> See shared response on detection baseline.
>
> ### Related Work
> We have added the suggested references and updated the related work. Thank you.

---

> > ### Comment · Reviewer_b7Zs · 2021-10-01
> > **Reply to authors**
> >
> > Dear authors,
> >
> > thanks for the detailed feedback. Most of my concerns are addressed from this rebuttals, and I would like to keep my acceptance position.
> >
> > Best,

---

### Official Review · Reviewer_UD5Z · 2021-09-18
**Impressive dataset collection while not much different from existing work**

**Rating:** 7
**Confidence:** 3
**Clarity:** The paper is well written and clear.

**Strengths:**

Much of the autonomous driving research community relies on large real-world datasets. Collecting and preparing such a dataset is an immense effort that is only attempted sparsely by large teams. Although this is not by itself a strength of the submission, it is certainly acknowledged by the reviewer.

1) New real-world data at scale. The datasets provide new real-world data with a diverse set of sensors and large-scale annotation in the case of object detection and tracking.
2) There are clear (although minor) advantages to other existing datasets. Such as HD maps for all of them but also some unique features like stereo camera (missing in waymo, nuscenes etc)
3) Special care is taken to incorporate "interesting" scenarios for motion forecasting that is not always the case for other related datasets.

**Weaknesses:**

1) Writing [related work]: The related work section does not make clear what is the difference of the proposed datasets to the related datasets. Instead, contrastive statements are spread in the introduction and the dataset section (section 3). As a result, it is more difficult to understand the uniqueness of the proposed datasets wrt related work.

2) Novelty in dataset design: Apart from the aforementioned strength, the design of the dataset is relatively close to the existing ones. The novelty aspect is mostly present in aspects such as diversity and scale.

3) Missing justification and explanation of why the proposed dataset is more challenging or an interesting addition wrt existing datasets.

4) Missing experiments for Sensor Dataset

**Additional Feedback:**

1) line: 73-74:
2) Table 4 (+ text): The definition of "social context" is not clear from the text.
3) line 299 (Limitations): "... and we would do so [correct egregious mistakes] for these datasets". Why was this not already done? It should be part of preparing and cleaning a dataset.
4) line 278: "significant increase in dataset difficulty when compared with AV 1.0". Could you explain why? It's not clear from the text and numbers.
5) line: 73, 74: is there groundtruth data for optical or scene flow?
6) preview2 (sensor dataset): the bounding boxes for the pillars seem to be off and/or inconsistent. Is this a data problem or a visualization problem?
7) Motion forecasting dataset: what is the main difference to the Waymo and Lyft datasets? Why should people use the proposed motion forecasting dataset? I understand that there is table 1 which illustrates this to some degree but it would help to explicitly state the important/key differences to motivate the usage of the dataset.

**Correctness:**

The claims seem to be correct and the dataset is constructed in a sound way.

What is missing are the evaluation metrics for the Sensor dataset.

**Documentation:**

There is no official documention yet. Based on Argoverse 1.0, this will be no concern.

**Ethics:**

No concerns

**Relation To Prior Work:**

See weakness section

**Summary And Contributions:**

Argoverse 2.0 consists of 3 driving datasets dubbed 'Sensor', 'Lidar' and 'Motion Forecasting'. The 'Sensor' dataset consists of 1000 scenes in 6 cities and focuses on 3D tracking. 'Lidar' is a large dataset specifically designed for self-supervised learning and point cloud forecasting. Finally, 'Motion Forecasting' is a next gen iteration on the Argoverse 1.0 motion forecasting dataset. What all of these datasets have in common is that they are more diverse than prior work and often also larger such that high-capacity models can be trained on them. Additionally, all three datasets are accompanied with HD maps to provide strong priors.
Finally, special care is taken to address the trade-off of dataset size and utility for the academic community without access to vast computational resources.

---

> ### Author Response · Authors · 2021-09-30
> **Response to R1**
>
> Thank you for your feedback. We greatly appreciate it and will use it to make both the paper and the datasets better.
>
> ### Related Work
>
> We have rewritten it to clarify the relationship between cited works and our proposed datasets.
>
> ### Novelty in dataset design: Apart from the aforementioned strength, the design of the dataset is relatively close to the existing ones. The novelty aspect is mostly present in aspects such as diversity and scale
>
> Good point.  There are several similarities between Argoverse 2.0 and other AV sensor and forecasting datasets.  Significantly, there are also important differences in dataset design and operating domains.  Due to the safety critical nature of perception, tracking, prediction, and planning for autonomous vehicles, it is essential that the research community has access to multiple high quality datasets.  This allows the community as a whole to better evaluate and understand the generalizability of new methods and to explore transfer learning.  We believe that impactful papers in this space should routinely publish results for multiple test sets.  Consequently, while not entirely novel, we believe that Argoverse 2.0 is likely the most widely useful dataset to date.  At a bare minimum it represents an important and well constructed addition to the small cannon of datasets on which to evaluate developments in this space.
>
> ###  “justification and explanation of why the proposed dataset is more challenging or an interesting addition wrt existing datasets.”
>
> See shared response concerning dataset comparisons.
>
> ### “Missing experiments for Sensor Dataset”
>
> See shared response concerning detection baseline.
>
>
> ### Regarding additional feedback
>
> ### Q2: Table 4 (+ text): The definition of "social context" is not clear from the text.
>
> We have clarified the definition of “social context” in the manuscript. By social context, we mean  encoding other actors’ states in the feature representation.
>
> ### Q4: "significant increase in dataset difficulty when compared with AV 1.0". Could you explain why? It's not clear from the text and numbers.
>
> We understand the lack of clarity there as the relevant text was moved to the supplementary material, instead of the main paper. Figure 2 in the Supplement shows that Argoverse 2.0 achieves a much higher interestingness score compared to Argoverse 1.1. The same plot also indicates that the interestingness score has a high correlation with Miss Rate metric for any particular motion forecasting method, thereby justifying that the interestingness score computed here is indeed a reliable measure of difficulty. We have moved this figure back to the main text now that an extra page is available. Finally, Table 2 in supplementary shows that transfer learning from 2.0 to 1.1 is more useful than the reverse despite having a lesser number of scenarios. This reinforces the fact that Argoverse 2.0 is more challenging than its predecessor.
>
>
> ### Q5: groundtruth data for optical or scene flow
>
> Sensor datasets with 3D cuboid annotations can be used as ground truth for scene flow algorithms, as was done in [Scene Flow from Point Clouds with or without Learning](https://arxiv.org/abs/2011.00320) from Argo and [Scalable Scene Flow from Point Clouds in the Real World](https://arxiv.org/abs/2103.01306) from Waymo. Projecting these flow vectors to 2D would create optical flow annotations. In both cases, the annotations can be imperfect, especially for articulated, deformable objects like pedestrians.
>
> ### Q6. Bollards in visualization 2.
>
> The second visualization did have room for improvement on the bollard cuboids. Thank you for spotting this. We confirmed that the cuboids properly contained the bollard lidar points, but we centered the cuboids more accurately, for this and all sequences.

---

> > ### Author Response · Authors · 2021-09-30
> > **Response to R1**
> >
> > ### Q7. Response to - Motion forecasting dataset: what is the main difference to the Waymo and Lyft datasets? Why should people use the proposed motion forecasting dataset?
> >
> > **Comparison to Waymo**: Argoverse 2.0 and Waymo are complementary datasets. There is significant value in having two similar high-quality datasets that come from totally non-overlapping distributions. That said, Argoverse has a much more accessible entry point - we’ve intentionally limited the scope of the dataset to ensure that the dataset is easy to distribute (our ~30GB vs their ~1.3TB).
> >
> > **Comparison to Argoverse 1.1**: This dataset was the first motion-forecasting specific dataset in the self-driving domain and was pivotal in influencing increased research activity in this domain. However, as evident in Figure 2 of supplementary material, the performance of forecasting methods has saturated, with no significant improvement in minFDE over the last several months. Furthermore, the lack of object categories and multi-agent evaluation, as well as shorter forecast horizon, and smaller quantity of challenging scenarios, has limited its use. Argoverse 2.0 overcomes all these shortcomings and provides a much more “complete” dataset to work with.
> >
> > **Comparison to Lyft**: Despite the fact that the Lyft dataset is the largest (measured by total log hours) on paper, the entire dataset was collected from a single 10 km stretch of road.
> >
> > **Comparison to nuScenes**: The nuScenes prediction dataset tackles a slightly different flavor of motion forecasting compared to Argoverse and Waymo. Observations are at a much lower frequency than almost every other dataset (2 Hz vs. 10 Hz). This means that some scenarios have very few observations (potentially as few as 1), making the prediction task quite difficult. This difficulty is reflected in the best ADE numbers reported on their leaderboard, which are significantly higher than both Argoverse and Waymo. Further, their prediction dataset is mainly a repurposed version of tracking dataset, as such it spans 5.5h compared to 305h of Argoverse 2.0. This means that there are going to be a large number of overlapping scenarios and not enough “new” observations to learn from.

---

> > > ### Comment · Reviewer_UD5Z · 2021-09-30
> > > **Response of R1**
> > >
> > > Thank you for the detailed feedback. It would help if you point to the supplementary material in the main text for the sake of clarity. Finally, the explanation in the rebuttal is quite clear and should be merged into the paper.

---

### Author Response · Authors · 2021-09-30
**Shared response to all reviewers on most frequently raised issues: dataset comparisons and detection baseline experiments.**

### Relationship to other datasets

We have added new comparisons to the manuscript to clarify why our datasets are “more challenging or an interesting addition w.r.t. existing datasets” (R1) and to add “justification for the necessity of the proposed dataset over existing ones” (R2). “there is no direct diversity comparison” (R3)

**Sensor Dataset**:  Figure 1, comparing the number of cuboids per category, is expanded to compare with ONCE and Waymo, in addition to nuScenes. ONCE and Waymo have smaller taxonomies. We expand Figure 2 left, showing the number of cuboids as a function of range, to include Waymo, ONCE, and nuScenes. This figure shows that we have dramatically more objects beyond 70 m in range than existing datasets. We also add a new figure, 2 right, which is a histogram over the number of objects present in each lidar frame. The average number of cuboids for Argoverse 2.0 is 75, compared to 33, 61, and 29 for nuScenes, Waymo, and ONCE, respectively. We include a new figure, 3 left, which is a histogram over the number of object categories simultaneously present in each lidar frame. This figure shows that our dataset has far more frames with 5+ categories of objects than existing datasets. The mode of this distribution is 8 objects for Argoverse 2.0, compared to 4 for nuScenes and ONCE, and 3 for Waymo. And while any expanded taxonomy is naturally heavy tailed, just as the real world is, we still have enough instances to evaluate detection and tracking for 26 different categories compared to 10 for nuScenes. Finally, we add Figure 3 right showing that our new sensor dataset has more moving vehicles than any other dataset.

**Motion Forecasting Dataset**: We expect Argoverse 2.0 to be more challenging and interesting than other relevant datasets. Since all the datasets have different raw format and source, it is non-trivial to compute the interestingness score for all of them using the exact same methodology. As such, we rely on some derived experiments and stats to compare the challenging/interesting nature with some of the other datasets (1) Table 2 shows that Argoverse 2.0 has the maximum number of object types. Methods now need to account for object type specific behaviors, e.g. BUS usually has a high turning radius, CYCLIST can cut through narrow spaces. It also shows that Argoverse 2.0 has the maximum scenario duration and maximum roadway coverage, which would make the forecasting methods learn from longer sequences and diverse road geometry. (2) We have updated the paper text to include Figure 6, which shows that Argoverse 2.0 achieves a much higher interestingness score compared to Argoverse 1.1.

The same plot also shows that the interestingness score has a high correlation with Miss Rate metric for a particular motion forecasting method, thereby justifying that the interestingness score computed here is indeed a reliable measure of difficulty. (3) Table 2 in supplementary shows that transfer learning from 2.0 to 1.1 is more useful than the reverse despite having a lesser number of scenarios. This reinforces the fact that Argoverse 2.0 is more challenging than its predecessor. (4) Finally, we compare the metric values between the state-of-art methods on the Waymo motion dataset and Argoverse 1.1. For a 3 second horizon on vehicle object types, the ones on Waymo achieve a minFDE of approximately 0.51 m. This is much lower compared to 1.13 m achieved on Argoverse 1.1. Further, the same method stands 1st on Waymo’s leaderboard (under the name Multipath++) and 5th on Argoverse’s (under the name poly).

We have added additional text to the related work section to more concretely position Argoverse 2.0 relative to other forecasting datasets.

**Lidar Dataset**: There is only one similar dataset, ONCE. Keep in mind that ONCE is concurrent work, released this summer and under submission to this same dataset track. Our lidar dataset consists of 6 million frames vs. 1 million for ONCE. ONCE samples are drawn at 2 hz instead of 10 hz for us, so the time period covered in each dataset is similar. However, our higher time resolution makes our dataset more suitable for tasks related to motion, e.g. the point cloud forecasting task we explore. Our dataset also contains HD maps for each scenario, while ONCE does not.

---

> ### Author Response · Authors · 2021-09-30
> **Shared response to all reviewers on most frequently raised issues: dataset comparisons and detection baseline experiments.**
>
> ### Baseline experiments for Sensor Dataset
>
> **3D Detection Baselines:** We have performed baseline 3D detection experiments on our proposed sensor dataset and added them to the manuscript. We use our own implementation of [CenterPoint](https://arxiv.org/abs/2006.11275) with a sparse VoxelNet backbone. Our implementation contains six detection heads to accommodate the larger taxonomy of our dataset and we trained the model using a learning rate of 3e-3 with the AdamW optimizer and the OneCycle learning rate scheduler. We processed the input point cloud at a voxel resolution of 0.1m, 0.1, 0.2m (xyz) with a 200m, 200m grid size. We train on the full training set and report results on the entire validation set. We plan to make our implementation publicly available.
>
> Figure 7 in the updated paper illustrates that this detector performs reasonably well on vehicles (AP .606) and pedestrians (AP .333), but performs poorly on several uncommon objects that are new to our taxonomy -- Dog AP .003, Stroller AP .014, Wheeled device (e.g. E-Scooter) AP .031. All of these classes have thousands of unique instances in the training split and 600+ unique instances (each instance typically has dozens of annotated frames) in the validation split, so it would be expected that with additional innovation, trained detectors will perform better on such classes.
>
> We evaluate our method using metrics from the Argoverse 3D Detection leaderboard which are largely analogous to nuScenes metrics: average precision (AP), average translation error, average scaling error (ASE), average orientation error (AOE), and composite detection score (CDS).
>
> | Class Names                         | AP    | ATE   | ASE   | AOE   | CDS   |
> |-------------------------------------|------:|------:|------:|------:|------:|
> | Regular Vehicle                     | 0.606 | 0.436 | 0.188 | 0.375 | 0.499 |
> | Pedestrian                          | 0.333 | 0.368 | 0.265 | 1.255 | 0.238 |
> | Bicyclist                           | 0.142 | 0.241 | 0.266 | 0.319 | 0.119 |
> | Motorcyclist                        | 0.160 | 0.286 | 0.288 | 0.427 | 0.130 |
> | Wheeled Rider                       | 0.000 | 2.000 | 1.000 | 3.142 | 0.000 |
> | Bollard                             | 0.246 | 0.357 | 0.504 | 0.831 | 0.168 |
> | Construction Cone                   | 0.085 | 0.406 | 0.481 | 0.497 | 0.061 |
> | Sign                                | 0.065 | 0.369 | 0.405 | 0.382 | 0.050 |
> | Construction Barrel                 | 0.226 | 0.355 | 0.419 | 0.572 | 0.167 |
> | Stop Sign                           | 0.284 | 0.315 | 0.433 | 0.243 | 0.220 |
> | Mobile Pedestrian Crossing Sign | 0.157 | 0.248 | 0.510 | 0.465 | 0.116 |
> | Large Vehicle                       | 0.022 | 0.600 | 0.279 | 0.478 | 0.017 |
> | Bus                                 | 0.364 | 0.526 | 0.196 | 0.273 | 0.298 |
> | Box Truck                           | 0.263 | 0.583 | 0.232 | 0.138 | 0.214 |
> | Truck                               | 0.143 | 0.522 | 0.175 | 0.170 | 0.120 |
> | Vehicular Trailer                   | 0.071 | 0.791 | 0.175 | 0.531 | 0.053 |
> | Truck Cab                           | 0.083 | 0.718 | 0.288 | 0.080 | 0.064 |
> | School Bus                          | 0.076 | 0.484 | 0.186 | 0.021 | 0.065 |
> | Articulated Bus                     | 0.085 | 0.846 | 0.171 | 0.127 | 0.067 |
> | Message Board Trailer               | 0.010 | 1.168 | 0.381 | 0.376 | 0.006 |
> | Bicycle                             | 0.095 | 0.463 | 0.280 | 0.627 | 0.072 |
> | Motorcycle                          | 0.128 | 0.293 | 0.265 | 0.712 | 0.101 |
> | Wheeled Device                      | 0.031 | 0.552 | 0.253 | 0.680 | 0.023 |
> | Wheelchair                          | 0.000 | 2.000 | 1.000 | 3.142 | 0.000 |
> | Stroller                            | 0.014 | 0.507 | 0.323 | 0.596 | 0.011 |
> | Dog                                 | 0.003 | 0.793 | 0.413 | 1.192 | 0.002 |
> | Average Metrics                     | 0.132 | 0.722 | 0.406 | 0.855 | 0.103 |

---

### Decision · Program_Chairs · 2021-10-09

**Decision:**

Accept

**Comment:**

The paper introduces three self-driving datasets: a multimodal dataset suitable for supervised perception tasks, a Lidar dataset that is the largest existent lidar dataset and suitable for self-supervised learning, and a rich motion forecasting dataset for prediction tasks, given history of object location, heading, velocity, and category. All the reviewers agreed that this is a good paper, providing a very diverse and useful resource for the self-driving community. The authors updated their submission to take into account reviewers' comments.